# Molecular basis of F-actin regulation and sarcomere assembly via myotilin

**Julius Kostan** [1], **Miha Pavšič** [2], **Vid Puž** [2], **Thomas C. Schwarz** [1], **Friedel Drepper** [3,4], **Sibylle Molt** [5], **Melissa Ann Graewert** [6], **Claudia Schreiner** [1], **Sara Sajko** [1], **Peter F. M. van der Ven** [5], **Adekunle Onipe** [1], **Dmitri I. Svergun** [6], **Bettina Warscheid** [3,4], **Robert Konrat** [1], **Dieter O. Fürst** [5], **Brigita Lenarčič** [2,7] *, **Kristina Djinović-Carugo** [1,2] *

**1** Department of Structural and Computational Biology, Max Perutz Labs, University of Vienna, Vienna, Austria, **2** Department of Chemistry and Biochemistry, Faculty of Chemistry and Chemical Technology, University of Ljubljana, Ljubljana, Slovenia, **3** Biochemistry and Functional Proteomics, Institute of Biology II, Faculty of Biology, University of Freiburg, Freiburg, Germany, **4** Signalling Research Centres BIOSS and CIBSS, University of Freiburg, Freiburg, Germany, **5** Institute for Cell Biology, Department of Molecular Cell Biology, University of Bonn, Bonn, Germany, **6** European Molecular Biology Laboratory, Hamburg Unit, c/o DESY, Hamburg, Germany, **7** Department of Biochemistry, Molecular and Structural Biology, Jožef Stefan Institute, Ljubljana, Slovenia

☯ These authors contributed equally to this work.
* Brigita.Lenarcic@fkkt.uni-lj.si (BL), kristina.djinovic@univie.ac.at (KD-C)

**Data Availability Statement:** The atomic coordinates have been deposited in the SASDB and are available with these links: SASDFZ7 Full-length myotilin with N-terminally fused thioredoxin tag https://www.sasbdb.org/data/SASDFZ7/

## Abstract

Sarcomeres, the basic contractile units of striated muscle cells, contain arrays of thin (actin) and thick (myosin) filaments that slide past each other during contraction. The Ig-like domain-containing protein myotilin provides structural integrity to Z-discs—the boundaries between adjacent sarcomeres. Myotilin binds to Z-disc components, including F-actin and α-actinin-2, but the molecular mechanism of binding and implications of these interactions on Z-disc integrity are still elusive. To illuminate them, we used a combination of small-angle X-ray scattering, cross-linking mass spectrometry, and biochemical and molecular biophysics approaches. We discovered that myotilin displays conformational ensembles in solution. We generated a structural model of the F-actin:myotilin complex that revealed how myotilin interacts with and stabilizes F-actin via its Ig-like domains and flanking regions. Mutant myotilin designed with impaired F-actin binding showed increased dynamics in cells. Structural analyses and competition assays uncovered that myotilin displaces tropomyosin from F-actin. Our findings suggest a novel role of myotilin as a co-organizer of Z-disc assembly and advance our mechanistic understanding of myotilin's structural role in Z-discs.

## Introduction

About 40% of the human body is comprised of skeletal muscle, whose contraction leads to locomotion [1]. The contractile machinery of cross-striated muscle cells is based on an impressive, almost crystalline array of thin (actin based) and thick (myosin based) filaments, arranged in repeating units, the sarcomeres. Stringent control of the precise layout of these filament systems is of utmost importance for efficient conversion of the force produced by the myosin–

4tm5723u5y/ SASDF28 Myotilin immunoglobulin domains Ig1Ig2 (220-452) https://www.sasbdb.org/data/SASDF28/b0sfyy2m3k/ SASDF38 Myotilin immunoglobulin domains Ig1Ig2 (250-444) https://www.sasbdb.org/data/SASDF38/knazhmxat5/ SASDF48 Myotilin immunoglobulin domains Ig1Ig2 (250-498) https://www.sasbdb.org/data/SASDF48/w8mwgpml0d/ SASDJH8 Myotilin immunoglobulin domains Ig1Ig2 (220-452, wild-type) concentration series data https://www.sasbdb.org/data/SASDJH8/6fkwyvfgcl/ SASDJJ8 Myotilin immunoglobulin domains Ig1Ig2 (220-452, R405K mutant) concentration series data https://www.sasbdb.org/data/SASDJJ8/af8sxzaqhi/.

**Funding:** This work was supported by the Slovenian Research Agency (arrs.si) young researcher grant (No. 35337) and research program P1-0140. KDC research was supported by a Marie Curie Initial Training Network (https://ec.europa.eu/research/mariecurieactions/funded-projects/initial-training-networks_en): MUZIC (N° 238423), Austrian Science Fund (FWF.ac.at) Projects I525, I1593, P22276, P19060 and W1221, Federal Ministry of Economy, Family and Youth through the initiative "Laura Bassi Centres of Expertise" (https://www.ffg.at/en/EN/program/laura-bassi-centres-expertise) funding the Centre of Optimized Structural Studies, N°253275, by the Wellcome Trust Collaborative Award (wellcome.org) (201543/Z/16), Austrian-Slovak Interreg (https://interreg.eu/programme/interreg-slovakia-austria) Project B301 StruBioMol, COST action (www.cost.eu) BM1405 - Non-globular proteins - from sequence to structure, function and application in molecular physiopathology (NGP-NET), WWTF (Vienna Science and Technology Fund; www.wwtf.at) Chemical Biology project LS17-008, and by the University of Vienna. This study was further supported by the Deutsche Forschungsgemeinschaft (DFG, German Research Foundation; www.dfg.de) FOR 1352 (projects P1, D.F.; and P4, B.W.) FOR 2743 (projects P6, D.F.; and P9, B.W.) Project ID 403222702/SFB 1381 (project Z1, B.W.) and Germany's Excellence Strategy (CIBSS – EXC-2189 – Project ID 390939984, BW). The FP7 WeNMR (www.wenmr.eu) (project# 261572), H2020 West-Life (github.com/h2020-westlife-eu) (project# 675858) and the EOSC-hub (www.eosc-hub.eu) (project #777536) European e-Infrastructure projects are acknowledged for the use of their web portals, which make use of the EGI infrastructure with the dedicated support of CESNET-MetaCloud, INFN-PADOVA, NCG-INGRID-PT, TW-NCHC, SURFsara and NIKHEF, and the additional support of the national GRID Initiatives of Belgium, France, Italy, Germany, the Netherlands, Poland, Portugal,

actin interaction into contraction at the macroscopic level. Thin filaments are cross-linked in an antiparallel fashion at the Z-discs, the boundaries of adjacent sarcomeres, by multiple molecular interactions (Fig 1A). In fact, more than 50 proteins may be associated with mature Z-discs, and they are regarded as one of the most complex macromolecular structures in biology. For decades, the Z-disc was believed to play a specific role only in sustaining myofibril architecture. This view has changed dramatically within the last decade, with the Z-disc now also being recognized as a prominent hub for signaling, mechanosensing, and mechanotransduction, with emerging roles in protein turnover and autophagy [2,3]. In line with this, Z-disc proteins have recently been identified as a major phosphorylation hotspot, thereby directly modulating protein interactions and dynamics [4,5].

A striking ultrastructural feature of the Z-disc is its highly ordered, paracrystalline tetragonal arrangement [7]. This arrangement is governed by α-actinin-2, which cross-links the overlapping ends of thin filaments of neighboring sarcomeres and provides structural integrity. α-Actinin-2 also binds to titin, which is connected to thick filaments. This interaction is regulated by a phosphoinositide-based mechanism [8,9]. However, many other details of the molecular architecture of the Z-disc remain elusive, and many questions still need to be addressed. For example, why is tropomyosin distributed all along thin filaments of the sarcomere with exception of the Z-disc [7]? How are interactions of α-actinin-2 with numerous binding partners translated into Z-disc assembly?

The Z-disc component myotilin is involved in multiple interactions by directly binding to α-actinin-2, filamin C, FATZ/myozenin/calsarcin, ZASP/cypher, and F-actin [10–14]. It was therefore identified as a key structural component of Z-discs and proposed to control sarcomere assembly [10–12]. Mutations in the human myotilin gene are associated with myofibrillar myopathy (MFM) [15,16], supporting the notion that myotilin is important for proper maintenance, organization, and/or function of the Z-disc.

Myotilin is a member of the palladin/myopalladin/myotilin family and contains 2 Ig domains that are involved in myotilin dimerization and interact with F-actin and filamin C (Fig 1B) [10,11,17,18]. The shortest fragment of myotilin that can bind F-actin includes the 2 Ig domains preceded by a stretch of intrinsically disordered residues (aa 214–442) [18]. A longer myotilin fragment including N-terminal and carboxyl-terminal regions flanking the 2 Ig domains (aa 185–498) can cross-link actin filaments in vitro and in vivo [18]. In addition, full-length protein prevents F-actin disassembly induced by latrunculin A [11], revealing its role in stabilization and anchoring of the thin filaments in the Z-discs [11]. Myotilin dimerization was proposed to be important for F-actin cross-linking [11]. The shortest myotilin fragment with the ability to dimerize includes the Ig2 domain and the carboxyl-terminal tail (aa 345–498) [18]. Although an antiparallel myotilin dimer was suggested [11], the insight into molecular determinants of myotilin dimerization remains limited.

To elucidate the molecular mechanism of myotilin interaction with F-actin and the role of this interaction in sarcomeric Z-disc assembly and structure, we used an integrative structural biology approach to build the first structural model of the myotilin:F-actin complex, which can serve as a blueprint for interaction of Ig domain-containing proteins with F-actin via 2 or more consecutive domains, such as the entire palladin family and filamins [19–22]. This structural model suggests that binding sites for myotilin, tropomyosin, and α-actinin-2 on F-actin partially overlap, as confirmed by competition assays. Furthermore, we characterized the interaction of myotilin with α-actinin-2 and showed that PI(4,5)P$_2$ does not have a direct regulatory effect on myotilin.

Based on our results, we propose a model in which myotilin simultaneously binds F-actin and α-actinin-2 with the concomitant displacement of tropomyosin. This renders myotilin not only structural support for Z-disc architecture but also a co-organizer of Z-disc assembly.

Spain, UK, Taiwan and the US Open Science Grid. The funders had no role in study design, data collection and analysis, decision to publish, or preparation of the manuscript.

**Competing interests:** The authors have declared that no competing interests exist.

**Abbreviations:** ABD, actin binding domain; ACTN2-WT, wild-type α-actinin-2; CAMD, calmodulin-like domain; DMEM, Dulbecco's Modified Eagle Medium; DMTMM, 4-(4,6-dimethoxy-1,3,5-triazin-2-yl)-4-methylmorpholiniumchloride; Doc2b, double C2-like domain-containing protein beta; DSF, differential scanning fluorimetry; EOM, ensemble optimization method; FDR, false discovery rate; FRAP, fluorescence recovery after photobleaching; FWHM, full width at half maximum; HRR, hydrophobic residues-rich region; HSQC, heteronuclear single quantum coherence; IDP, intrinsically disordered protein; IDR, intrinsically disordered region; ITC, isothermal titration calorimetry; MFM, myofibrillar myopathy; mgf, Mascot generic format; MST, microscale thermophoresis; NMR, nuclear magnetic resonance; NSD, normalized spatial discrepancy; POPC, 1-palmitoyl-2-oleoyl-sn-glycero-3-phosphocholine; ppm, parts per million; PTM, posttranslational modification; RALLS, right-angle laser light scattering; RI, refractive index; ROI, region of interest; SEC, size exclusion chromatography; SEC-SAXS, size exclusion chromatography coupled to SAXS; SRR, serine-rich region; TDA, triple detector array; Trx, thioredoxin; XL-MS, cross-linking coupled to mass spectrometry.

# Results

## Myotilin is a conformational ensemble in solution

The previously performed disorder tendency analysis of full-length myotilin revealed that the N-terminal region of myotilin displays characteristics of an intrinsically disordered protein (IDP) [23]. The meta-structure analysis, which reflects how likely a specific residue is to be located within a compact 3D structure, was carried out as described previously [6], and indicated a tendency for local compactions in the N-terminal ID region (Fig 1B).

For characterization of full-length myotilin, we fused thioredoxin (Trx) to its N-terminus (Trx-MYOT) to prevent progressive N-terminal degradation. A globular protein at the N-terminus of myotilin was shown not to disturb its function in vivo, as demonstrated by extensive use of GST, Myc, and GFP fusions in cell biophysics experiments [14,24,25]. To investigate the hydrodynamic and structural properties of Trx-MYOT, we first performed size exclusion chromatography coupled to SAXS (SEC-SAXS) (S1A Fig). Right-angle laser light scattering (RALLS) data collected in parallel confirmed that the elution peak corresponded to a monomeric protein (S1A Fig). The overall molecular parameters (molecular mass, the radius of gyration $R_g$, and maximal intramolecular distance $D_{max}$) derived from the resulting SAXS profile are also compatible with a monomeric species (Fig 1C, S1 Table). The featureless descent of the SAXS profile in log plot (Fig 1C) and the plateau in the dimensionless Kratky plot (S1B Fig) are characteristics for the scattering of particles that are disordered and/or flexible. The calculated atom-pair distance distribution functions $P(r)$ skewed to shorter distances, and the $D_{max}$ value of approximately 200 Å also suggest flexible and elongated "string like" structures (Fig 1C, inset).

The derived ab initio molecular envelope of Trx-MYOT displayed an elongated multi-sub-unit shape (Fig 1D). To better account for the flexibility of Trx-MYOT, we employed the ensemble optimization method (EOM) [26]. A genetic algorithm-based selection of representative models with their respective volume fractions in solution resulted in the best fit to the data with $\chi^2 = 1.01$ (Fig 1C and 1E). Overall, the EOM-selected models tended to be more compact than those in the random pool. This is visible from the shift to lower values of $D_{max}$ and $R_g$ derived from the selected models compared to the distribution derived from the original pool, as the most extended models were not selected (Fig 1E, S1C and S1D Fig). Based on this analysis, Trx-MYOT displays a tendency to occupy more compact conformations compared to a chain with random coil linkers, in line with the predicted compactness (Fig 1B) and the disorder tendency analysis [23].

In summary, Trx-MYOT in solution displays a flexible conformational ensemble with local clusters of compactness in the intrinsically disordered regions (IDRs), in addition to the structured Ig domains connected by flexible linkers.

## Myotilin forms concentration-dependent dimers

Although myotilin was suggested to form antiparallel dimers [11], full-length myotilin was monomeric under our experimental conditions. The shortest myotilin fragment with the ability to dimerize was reported to include the Ig2 domain and the carboxyl-terminal tail [18]. To investigate molecular determinants of dimerization, we used SAXS on a series of truncated constructs—Ig1Ig2$^{250–444}$ (Ig1Ig2 tandem), Ig1Ig2$^{220–452}$, and Ig1Ig2$^{250–498}$—due to the inherently low solubility of Trx-MYOT.

We monitored the maximum scattering intensities extrapolated to zero angle ($I_0$) for a concentration series of the Ig1Ig2$^{220–452}$ construct and observed a concentration-dependent increase in the average molecular mass of resulting particles (Fig 1F, inset). Accordingly, the $P$

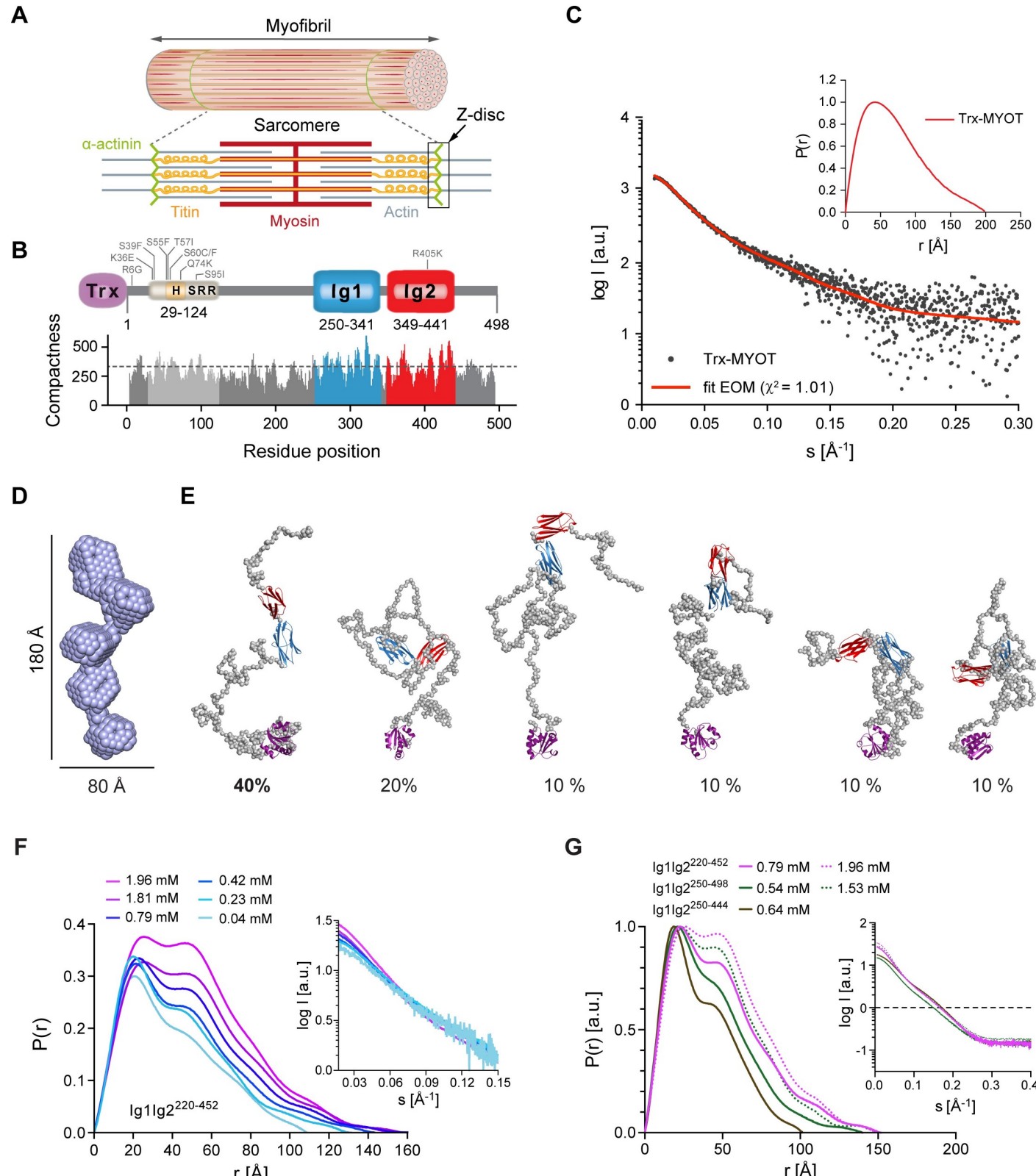

**Fig 1. Myotilin displays conformational ensemble in solution.** (A) Schematic representation of striated muscle sarcomere with its major thick (myosin based) and thin (actin based) filaments, titin, and the Z-disc anchoring cross-linker α-actinin. Extending from one Z-disc to the next, sarcomere represents the fundamental unit

of muscle contraction. **(B)** Schematic presentation of myotilin and its predicted compactness as a function of residue position. The N-terminal part of myotilin contains the SRR, which comprise hydrophobic residues stretch (H, yellow) and represents "mutational hotspot" of the protein (gray). Known disease-causing mutations are shown. Ig domains 1 and 2 are colored blue and red, respectively, followed by the carboxyl-terminal tail. Within this study, full-length myotilin construct fused to N-terminal Trx (violet) was used. Large meta-structure derived compactness values indicate residue positions typically buried in the interior of the 3D structure, whereas small values are found for residues exposed to the solvent. Dashed line depicts the average compactness value (about 300) of the well-folded protein. Significantly smaller values (200) are found for structurally flexible proteins [6]. **(C)** Experimental SAXS scattering data of Trx-MYOT, with the fit of the EOM flexible modeling. Inset, $P(r)$ vs. $r$ plot for Trx-MYOT with the $D_{max}$ ~200 Å. **(D)** SAXS-based ab initio molecular envelope of Trx-MYOT. Most probable model is shown. **(E)** EOM models of Trx-MYOT showing flexible and intrinsically disordered N-terminal and carboxyl-terminal regions (gray). The selected models are presented with the percentage contribution, estimated from the final population of EOM models. **(F)** $P(r)$ vs. $r$ plot for concentration series of Ig1Ig2$^{220-452}$. Inset, respective concentration series with the corresponding SAXS profiles. **(G)** $P(r)$ vs. $r$ plot for dimeric Ig1Ig2$^{220-452}$, Ig1Ig2$^{250-498}$, and monomeric Ig1Ig2$^{220-444}$. Inset, SAXS profiles of respective constructs. In order to compare various $P(r)$ functions, $P(r)$ was normalized to the peak height. Data points that were used to create graphs are reported in S2 Data. See also S1 Fig and S1 Table. The 3D models presented in this panel are available in the following links: https://skfb.ly/6YGyH and https://skfb.ly/6YGzt. EOM, ensemble optimization method; SRR, serine-rich region; Trx, thioredoxin.

$(r)$ function for Ig1Ig2$^{220-452}$ showed a transition from a distribution typical for an extended 2-domain protein, with a significant peak at approximately 25 Å and a minor at 50 Å, to a distribution with increased frequencies of the 50 Å peak and a new minor signal at in the region 80 to 100 Å (Fig 1F). Here, the shortest vector corresponds to the distances within Ig domains, while the second and the third peaks correspond to inter-domain distances within and between the subunits with concomitant increase of $D_{max}$ value to approximately 150 Å (Fig 1F, S1 Table). The notable increase of the $D_{max}$ value in a concentration-dependent manner from 110 Å to 150 Å together with aforementioned features of the $P(r)$ suggests an antiparallel staggered dimer formed either by interactions via Ig1 (head to head) or via Ig2 domains (tail to tail) or a staggered parallel dimer via Ig1-Ig2. The calculated structural parameters ($D_{max}$ and $P(r)$) of the non-staggered parallel or antiparallel dimer are not in line with the experimentally derived values, thus making such architecture less likely (Fig 1F, S1E and S1F Fig).

$P(r)$ functions of Ig1Ig2$^{250-444}$, Ig1Ig2$^{220-452}$, and Ig1Ig2$^{250-498}$ derived from scattering data at comparable concentrations showed that Ig1Ig2$^{220-452}$ and Ig1Ig2$^{250-498}$ displayed a high second peak at 50 Å and corresponding $D_{max}$ values of approximately 150 Å (Fig 1G, S1 Table), whereas Ig1Ig2$^{250-444}$ did not. Guinier region analysis revealed an apparent increase of $R_g$ for the Ig1Ig2$^{220-452}$ and Ig1Ig2$^{250-498}$, whereas Ig1Ig2$^{250-444}$ displayed molecular parameters of a monomeric species, comparable to those observed for Ig1Ig2$^{220-452}$ at low concentrations (Fig 1F, S1 Table). Consequently, molecular mass values calculated from $I_0$ using the Guinier approximation correspond to the expected molecular mass of a dimer for Ig1Ig2$^{220-452}$, approaching dimer for Ig1Ig2$^{250-498}$, and of a monomer for Ig1Ig2$^{250-444}$ (S1 Table), corroborating the role of regions flanking the Ig1Ig2.

R405 is mutated in muscular dystrophy and was suggested to be responsible for defective homodimerization of myotilin [27]. We therefore assessed the dimerization propensity of Ig1Ig2$^{220-452\ R405K}$ mutant using SAXS. Comparative analysis of SAXS data and derived molecular parameters ($D_{max}$, $R_g$, $P(r)$, molecular mass) revealed that the conserved R405K mutation which preserves the positive charge does not impair dimer formation in vitro (S1G Fig, S1 Table), in contrast to previously reported defective homodimerization observed by immunoprecipitation and yeast 2-hybrid screens [27].

Our results together with published data on myotilin dimerization collectively show that (i) N-terminal and carboxyl-terminal regions flanking the Ig1Ig2 domains contribute to dimer stabilization in solution; and (ii) the fully dimeric form is observed only at high protein concentrations (>1.0 mM), suggesting a weak association constant, explaining why only monomers of Trx-MYOT were observed in our SEC-SAXS experiments, where only low concentrations of Trx-MYOT could be used due to its aforementioned low solubility. Although tail-to-tail dimerization reconciles both SAXS analysis and proposed mode of incorporation in the Z-disc, orchestrated by interactions with filamin C and α-actinin-2 [11],

further higher-resolution structural studies are needed to describe the quaternary structure of myotilin in detail.

## Tandem Ig domains of myotilin together with contiguous regions are required for high-affinity binding to F-actin

To characterize myotilin–F-actin interaction and to elucidate the suggested role of segments flanking Ig1Ig2 [11,18], we determined the binding affinity of full-length Trx-MYOT, and a series of truncated constructs, using F-actin co-sedimentation assays (Fig 2A).

In the initial experiments (using conditions B1; for details, see Materials and methods) with single Ig domains (Ig1$^{250-344}$ and Ig2$^{349-459}$), and their tandem (Ig1Ig2$^{250-444}$), neither Ig1 nor Ig2 alone displayed significant binding to F-actin, while Ig1Ig2$^{250-444}$ bound to F-actin with $K_d$ = 9.2 ± 1.0 μM (Fig 2A, S2A Fig). When the same experimental conditions were used for Trx-MYOT, its self-pelleting was observed, making interpretation of the data difficult. To reduce the fraction of self-pelleted Trx-MYOT, we optimized the conditions for the co-sedimentation assays using a differential scanning fluorimetry (DSF)-based pH screen (S2C Fig). We found that a slightly acidic pH, i.e., an adjustment of the assay buffer from pH 7.4 (in conditions B1) to pH 6.8 (conditions B2; for details, see Materials and methods), rendered Trx-MYOT more stable as well as more soluble. To assess the effect of pH (6.8), the affinity of Ig1Ig2$^{250-444}$ to F-actin was determined using the conditions B2, too. The observed apparent $K_d$ (12.9 ± 1.5 μM) was in good agreement with the affinity measured in the B1 conditions ($K_d$ = 9.2 ± 1.0 μM) (Fig 2A, S2B Fig), suggesting that diverse conditions used do not have a major effect on the myotilin affinity to F-actin.

The apparent $K_d$ of Trx-MYOT for F-actin under conditions B2 is 0.29 ± 0.06 μM (Fig 2A, S2B Fig), indicating relatively strong binding compared to other F-actin binding proteins of the palladin family. For instance, the affinity of full-length palladin to F-actin was reported to be 2.1 ± 0.5 μM [19]. To further delineate the role of regions flanking Ig1Ig2, Ig1Ig2$^{185-454}$, Ig1Ig2$^{250-498}$, and Ig1Ig2$^{185-498}$ were assayed for F-actin binding (Fig 2A, S2B Fig). All constructs bound to F-actin with similar affinity ($K_d$ = 3.0 ± 0.6 μM, 1.2 ± 0.2 μM, and 2.0 ± 0.4 μM for Ig1Ig2$^{185-454}$, Ig1Ig2$^{250-498}$, and Ig1Ig2$^{185-498}$, respectively), and thus significantly stronger than the Ig1Ig2$^{250-444}$ encompassing only the 2 Ig domains (Fig 2A, S2B Fig).

The surface electrostatic potential of the tandem Ig1Ig2 domains displays positive clusters on one face of the domains ("BED" β-sheet) and in general a more pronounced basic character of Ig2 in contrast to Ig1 (Fig 2B). Positive clusters have also been observed in several other F-actin binding proteins, including palladin [29], suggesting an electrostatically driven interaction. To investigate the influence of ionic strength on the affinity, we performed co-sedimentation assays with increasing NaCl concentrations. As shown in Fig 2C, higher salt concentrations reduced the ability of myotilin to co-sediment with F-actin.

To further corroborate the electrostatic nature of the interaction, we gradually neutralized the positive charge by mutating the basic residues of the Ig2 domain to Ala in the construct Ig1Ig2$^{250-444}$. While mutations of Q356A, K411A, and K367A did not significantly affect binding to F-actin compared to the wild type, single mutations K358A, K359A, and K354A had a negative effect, which was remarkably potentiated in the double K354/359A mutant, and even more in the K354/358/359A triple mutant (Fig 2D). The latter mutant showed a 70% reduction in binding to F-actin, suggesting that this basic patch in Ig2 plays an important role in F-actin binding (Fig 2B and 2D).

The muscular dystrophy–causing mutant R405K (Ig1Ig2$^{R405K}$) [27] was also tested, but did not show any notable difference in F-actin affinity with respect to the wild type (Fig 2A, S2A Fig), which is in agreement with its lateral location with respect to the identified positively

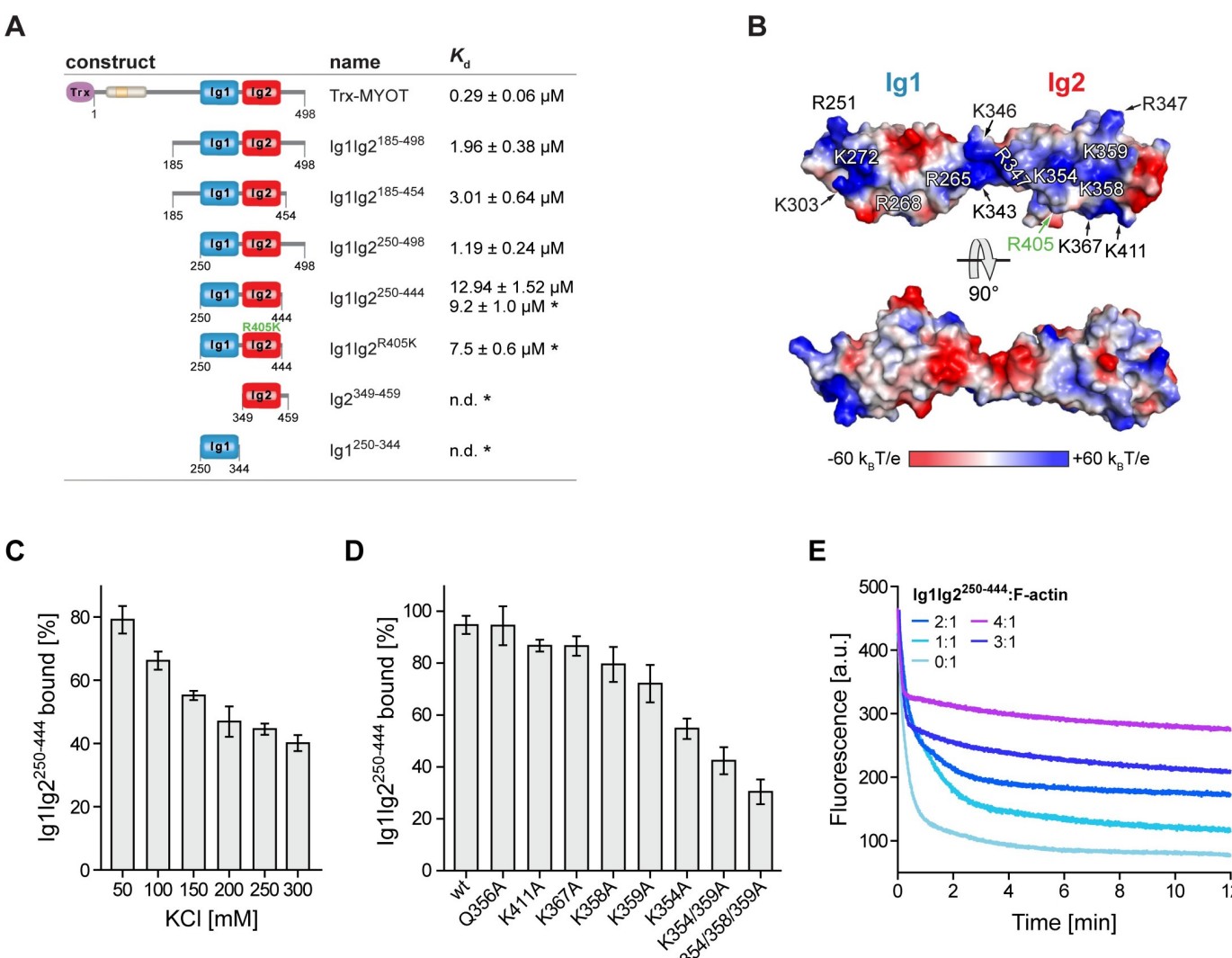

**Fig 2. Myotilin influences F-actin dynamics by binding via its Ig domains and disordered flanking regions. (A)** Summarized results of the actin co-sedimentation assays. The asterisk denotes values obtained in assays performed under B1 conditions (for details, see Materials and methods). **(B)** Surface electrostatic potential mapped on the Ig1Ig2 structural model [23]. Surfaces are colored on a red white blue gradient, as calculated by Adaptive Poisson-Boltzmann Solver [28]. Residues forming a basic patch (blue) on both Ig domains are labeled. The residue (R405) involved in muscular dystrophy–causing mutation R405K is shown in green [27]. **(C)** Binding of Ig1Ig2$^{250-444}$ to F-actin at increasing salt conditions. Graph data are presented as a percentage of theoretical total binding (100%) corresponding to the total amount of protein (8 μM) used in each experiment. Data represent mean values ± SEM of 3 independent experiments. The mean binding of Ig1Ig2$^{250-444}$ to F-actin in 50 mM KCl was significantly different to the means for all other tested concentrations of KCl. Significance was assessed using 1-way ANOVA with Holm–Sidak test, $p < 0.001$ for salt concentration of 50 mM vs. all other tested concentrations. **(D)** Binding of Ig1Ig2$^{250-444}$ and its mutant versions to F-actin. Graph data are presented as in (C). Data represent mean values ± SEM of 3 independent experiments. The mean binding to F-actin of all mutants except Q356A, K411A, and K367A were significantly different from the means of WT Ig1Ig2$^{250-444}$. Significance was assessed using 1-way ANOVA with Holm–Sidak test, $p < 0.01$ for WT vs. K358A and $p < 0.001$ for WT vs. all other mutants. **(E)** Effects of Ig1Ig2$^{250-444}$ on F-actin depolymerization. Actin filaments were depolymerized in the presence or absence of Ig1Ig2$^{250-444}$ at molar ratios indicated in the figure. Data points that were used to create graphs are reported in S2 Data. See also S2 Fig. The 3D model presented in this panel is available in the following link: https://skfb.ly/6YGzN. Trx, thioredoxin; WT, wild-type.

charged patch involved in F-actin binding. Furthermore, the mutation retains the positive charge at this position (Fig 2B).

Since binding data are available for other actin-binding Ig domains, e.g., for Ig domains of palladin and filamin A [20], we conducted a comparative structural analysis of the actin-binding Ig domains of myotilin, the Ig3 domain of palladin, and Ig10 of filamin A, which revealed

charged residues positioned on the same face in all analyzed Ig domains (S2D Fig). Notably, these residues coincide with those indicated by coevolution analysis [30] as potentially functional sites (S2D Fig, lower panel). Specifically, lysine residues K1008, K1011, and K1044 on palladin Ig3 domain were shown to be essential for F-actin binding [29]. Equivalent residues are also present in myotilin (S2D Fig, upper panel) and form a basic patch, which extends from one Ig domain to the other (Fig 2B).

Finally, we biochemically characterized the previously observed myotilin binding to (monomeric) G-actin [18] using a constitutively monomeric mutant of actin, DVD-actin [31]. We employed microscale thermophoresis (MST) with various myotilin tandem or isolated Ig domain constructs. The affinity of myotilin constructs to non-polymerizable DVD-actin was by 2 orders of magnitude lower than that to F-actin (S2E and S2F Fig). The highest affinity displayed by construct Ig1Ig2$^{185-454}$ indicates that, similarly to F-actin, the Ig domain flanking regions could be involved in modulating actin dynamics.

We hence performed depolymerization assays, to obtain further insight into the role of myotilin in actin dynamics. As also observed for the full-length myotilin [11], addition of Ig1Ig2$^{250-444}$ reduced F-actin depolymerization rate in a concentration-dependent manner (Fig 2E), suggesting that the tandem Ig domains play a critical role in actin binding.

In summary, our results reveal that myotilin has a role in stabilization of actin filaments and actin dynamics, where the tandem Ig domains play the central role and that both the N-terminal and carboxyl-terminal neighboring regions further enhance affinity and stabilize the myotilin–F-actin interaction (Fig 2A).

## Integrative structural model of the myotilin:F-actin complex

To validate and identify potential new contacts of F-actin with myotilin Ig domains and their contiguous regions, as implied from the binding and mutational data (Fig 2A and 2D), we performed cross-linking coupled to mass spectrometry (XL-MS) experiments on Ig1Ig2$^{250-444}$, Ig1Ig2$^{185-454}$, and Ig1Ig2$^{185-498}$ bound to F-actin (S3A–S3C Fig). As expected, specific cross-links involving both Ig1 and Ig2 domains as well as their N-terminal and carboxyl-terminal flanking regions (Fig 3, S2 Table) were found. While only 1 cross-link was detected between the Ig1 domain and F-actin (K303 on Ig1 and D27 on actin), there were several cross-links between the Ig2 domain and F-actin. This is in line with its higher affinity and the charged nature of this interaction (Fig 2B and 2D, S2A Fig), leading to cross-links of basic lysine residues on Ig2 domain with the acidic residues located in subdomain 1 of actin. Of the Ig2 domain residues cross-linked to F-actin, we also found K354 involved in the interaction using mutagenesis and co-sedimentation assays (Figs 2D and 3). Furthermore, the cross-links between acidic myotilin residues (D236, D239, and D241) and basic K328 and K330 on actin (Fig 3, S2 Table) map to the region preceding myotilin Ig1 in constructs Ig1Ig2$^{185-454}$ and Ig1Ig2$^{185-498}$. The latter construct yielded additional cross-links between positively charged K452, K462, K469, and K474 mapping to the region carboxyl-terminal to Ig2 and negatively charged residues E336, D27, and D26 on actin (Fig 3, S2 Table).

To construct an integrative molecular model of the myotilin:F-actin complex, we combined our structural model of Trx-MYOT (Fig 1E) and experimental data derived from in vitro mutational analysis and XL-MS, which we used as experimental distance restraints that guided the macromolecular docking using Haddock 2.2 [32] (for details, see Materials and methods). In this model the tandem Ig domains land on the subdomains 1 of the adjacent actin subunits, with the flexible N-terminal and carboxyl-terminal regions extending to the binding sites on neighboring actin subunits as suggested by XL-MS data (S3D Fig), supporting and explaining the published data [11,18].

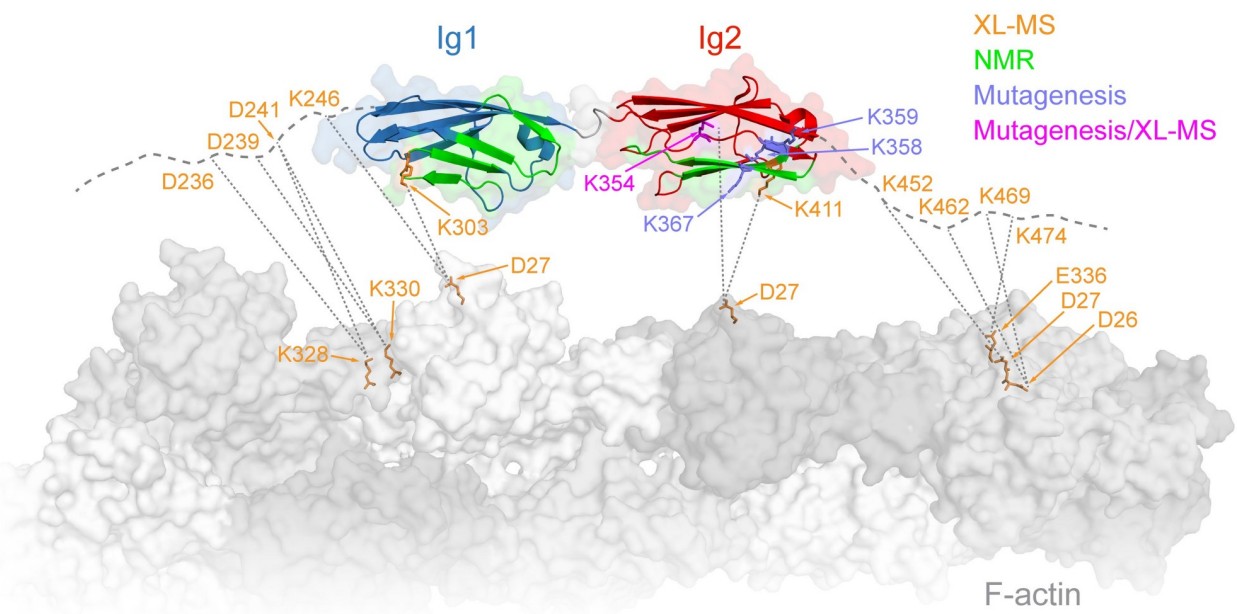

**Fig 3. Integrative structural model of myotilin:F-actin complex.** Summary of experimental data on myotilin:F-actin complex, obtained from XL-MS (orange), mutagenesis/co-sedimentation analysis (medium slate blue), and NMR (green). K354 was found to be involved in the interaction by both XL-MS and mutagenesis/co-sedimentation analysis (magenta). Thick, gray dashed lines represent N-terminal and carboxyl-terminal flanking regions of myotilin Ig domains. Selected residues found in cross-links between actin and myotilin are shown in orange. NMR, nuclear magnetic resonance; XL-MS, cross-linking coupled to mass spectrometry.

Nuclear magnetic resonance (NMR) was subsequently used to validate the structural model of myotilin:F-actin complex. NMR spectra of the isolated $^{15}$N-labeled Ig1 and Ig2 domains showed similar characteristics to those available in the BMRB databank (Ig1: 7113; Ig2: 16370) [33], and hence de novo assignment was not required in order to obtain a subset of assigned peaks. Construct Ig1Ig2$^{250-444}$ also showed a similar peak pattern, allowing the use of assignments of the single domains to obtain a partial assignment of this tandem construct. In the initial heteronuclear single quantum coherence (HSQC) experiments, $^{15}$N-labeled Ig1$^{250-344}$ was titrated with F-actin, which induced a reduction of the signal intensities (S3E Fig). Measurements of the $^{15}$N-labeled Ig2 domain (Ig2$^{349-459}$) showed a stronger reduction in signal intensity upon addition of F-actin compared to those carried out with the Ig1 domain (S3E Fig), indicating stronger binding of Ig2 to F-actin.

Further experiments with $^{15}$N-labeled Ig1Ig2$^{250-444}$ showed a distinctive shift pattern upon addition of F-actin, where the regions with significant HSQC shifts are 257 to 270, 296 to 309, 363 to 374, and 405 to 413 (S3F and S3G Fig) with segment 301 to 306 unfortunately not being represented due to low signal intensity [33]. These regions map to the "BED" β-sheet of both Ig domains and additionally to the β-strand A' in Ig1 and are in agreement with the interaction sites mapped by binding, mutational, and XL-MS data (Figs 2A and 2D and 3).

## Myotilin competes with tropomyosin and α-actinin-2 for the binding sites on F-actin

The actin residues identified in the myotilin–F-actin interaction by XL-MS form 2 clusters, a basic one comprised of K328 and K330 and an acidic one comprised of D26, D27, and E336 (Fig 3, S3D Fig). Interestingly, K328, together with neighboring K326, forms the major interaction site for tropomyosin on F-actin [34], implying that myotilin and tropomyosin could

compete for binding to F-actin. We therefore created a structural model of myotilin bound to F-actin in the presence of tropomyosin (Fig 4A). In this model, 2 Ig domains of myotilin only marginally overlap with tropomyosin bound to F-actin. However, their N-terminal and carboxyl-terminal flanking regions found by XL-MS analysis to interact with actin residues (D26, D27, K326, K328, and E336), map to or close to the tropomyosin binding site (Fig 4A), suggesting that they might interfere with F-actin–tropomyosin binding.

To test this hypothesis, we performed competition co-sedimentation assays where F-actin was incubated with increasing concentrations of Trx-MYOT, either before or after adding a fixed amount of the striated muscle isoform of human tropomyosin (Tpm1.1). The amount of tropomyosin bound to F-actin decreased with increasing concentrations of Trx-MYOT (Fig 4B, S4A Fig), uncovering that myotilin competes with tropomyosin for F-actin binding and displaces tropomyosin from F-actin.

In the myofibrillar Z-disc, myotilin associates with the striated muscle-specific isoform α-actinin-2 [10,12], the major Z-disc protein. As the stress fiber–associated tropomyosin isoforms were shown to compete with non-muscle α-actinin-1 and α-actinin-4 for F-actin binding [36,37], we examined whether α-actinin-2 is also able to compete with tropomyosin for F-actin. We used the constitutively open variant of α-actinin-2, in which mutations abrogating binding of EF34 to the α-actinin-2 "neck" region were introduced (ACTN2-NEECK, Fig 4C) to resemble the Z-disc bound conformation of α-actinin-2 [8]. We performed competition co-sedimentation assays, in which α-actinin-2 was incubated at increasing concentrations with F-actin before the addition of tropomyosin. As expected, binding of α-actinin-2 to F-actin prevented tropomyosin-F-actin binding in a concentration-dependent manner (Fig 4D, S4B Fig).

Comparative structural analysis of the myotilin:F-actin:tropomyosin model (Fig 4A) and the cryo-EM structures of F-actin decorated with the actin binding domain (ABD) of α-actinin-2, spectrin, or filamin [38–40] showed that binding sites of myotilin, tropomyosin, and α-actinin-2 on F-actin overlap (S4C Fig). Furthermore, comparison of binding affinities of myotilin, tropomyosin, and α-actinin-2 to F-actin [36,41–44], together with our competition assays (Fig 4B and 4D), indicate that myotilin could displace both tropomyosin and α-actinin-2 from F-actin, suggesting a novel role of myotilin as a regulator of access of tropomyosin and other actin-binding proteins to F-actin in the Z-disc.

## Myotilin binds α-actinin-2 using the same pseudoligand regulatory mechanism as titin

α-Actinin-2 interacts with titin as well as with palladin family of proteins via its carboxyl-terminal EF-hands (EF34) of the calmodulin-like domain (CAMD) [45–47]. The interaction between α-actinin-2 and titin binding motifs, the Z-repeats, has been well characterized [8,9,45] and was shown to be regulated by an intramolecular pseudoligand mechanism, in which a titin Z-repeat-like sequence ("neck," which contains the 1-4-5-8 binding motif) connecting the ABD and the first spectrin repeat of α-actinin-2, interacts with EF34 of the juxtaposed CAMD (Fig 4C). In this closed conformation, EF34 are not available for interactions to titin Z-repeats unless PI(4,5)P$_2$ catalyzes conformational switch to the open confirmation through their release from the "neck," and hence activating titin binding [8].

The interaction between α-actinin-2 and myotilin maps to the N-terminal part of myotilin, which also contains the conserved 1-4-5-8 motif (residues 95 to 106, Fig 4C and 4E) similar to palladin and myopalladin [47]. This was suggested by NMR experiments where a myotilin peptide containing the binding motif (Fig 4E) was titrated to EF34 [47].

To characterize the interaction of myotilin with α-actinin-2 using full-length proteins, we first performed pull-down assays using either wild-type α-actinin-2 (ACTN2-WT), its

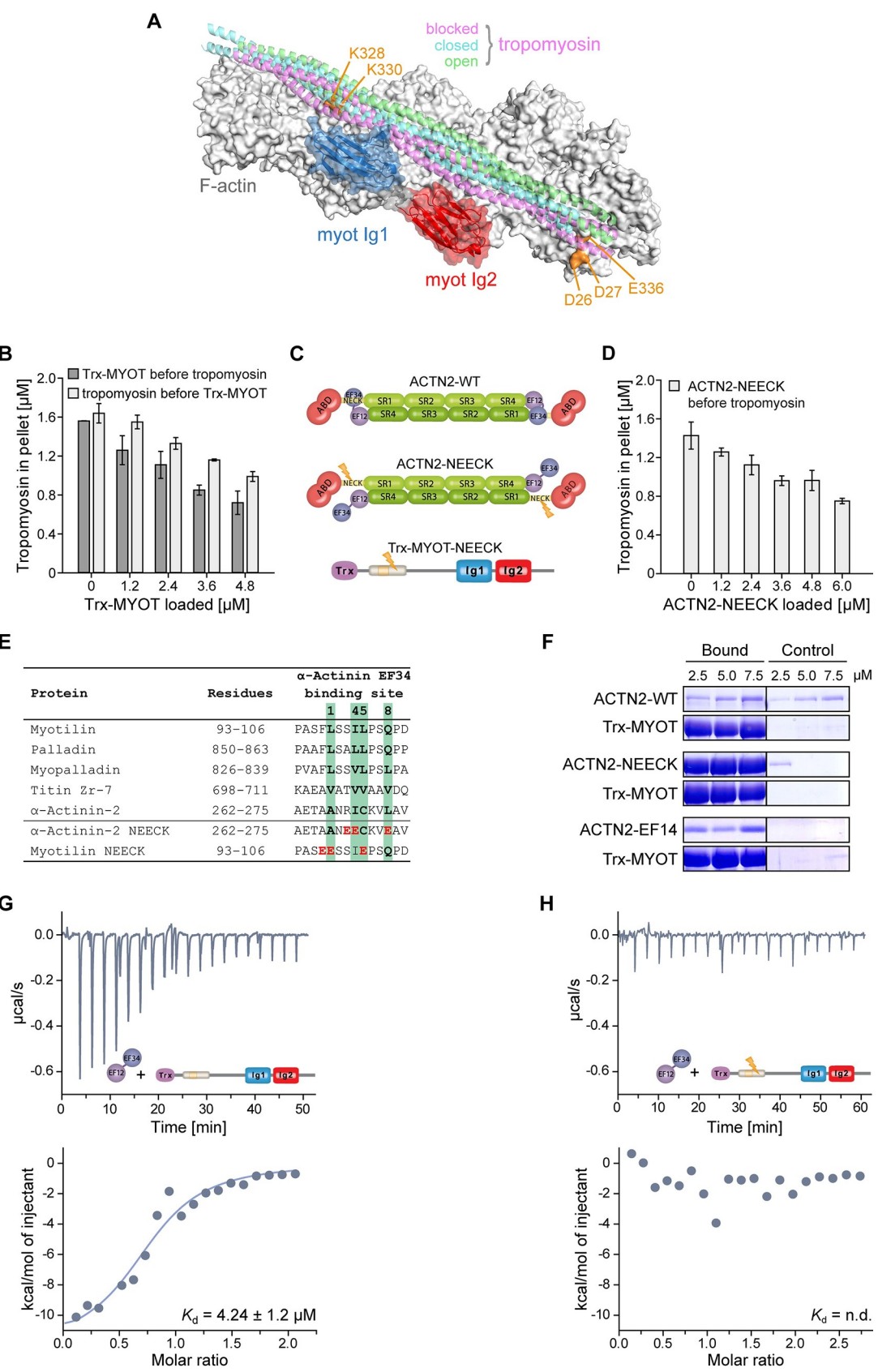

**Fig 4. Myotilin regulates binding of tropomyosin to F-actin and interacts with α-actinin-2. (A)** Model of tandem Ig domains of myotilin bound to F-actin superimposed with the cryo-EM structures of cardiac tropomyosin bound to F-actin in the blocked (magenta ribbons, PDB: 5NOG), closed (cyan ribbons, PDB: 5NOL), and open (green ribbons, PDB: 5NOJ) structural states [35]. Residues of actin found in cross-links with N-terminal and carboxyl-terminal regions flanking myotilin Ig domains are shown in orange (see Fig 3). **(B)** Effects of myotilin on tropomyosin:F-actin interaction. Myotilin (Trx-MYOT) at increasing concentrations was added to F-actin before (Trx-MYOT before tropomyosin) or after (tropomyosin before Trx-MYOT) incubation of F-actin with the fixed amount of tropomyosin. Mean values (± SEM) of 3 independent experiments are shown. For "Trx-MYOT before tropomyosin," the mean binding of tropomyosin to F-actin at all concentrations of Trx-MYOT was significantly different from the means in the absence of Trx-MYOT. Significance was assessed using 1-way ANOVA with Holm–Sidak test, $p < 0.01$ for 0 vs. 1.2 μM Trx-MYOT and $p < 0.001$ for 0 vs. all other concentrations of Trx-MYOT. For "Tropomyosin before Trx-MYOT," the mean binding of tropomyosin to F-actin at all concentrations of Trx-MYOT, except for 1.2 μM Trx-MYOT, was significantly different from the means in the absence of Trx-MYOT. Significance was assessed using 1-way ANOVA with Holm–Sidak test, $p < 0.05$ for 0 vs. 2.4 μM Trx-MYOT, $p < 0.01$ for 0 vs. 3.6 μM Trx-MYOT, and $p < 0.001$ for 0 vs. 4.8 μM Trx-MYOT. **(C)** Schematic presentation of the ACTN2-WT, its constitutively open mutant (ACTN2-NEECK), and mutant of Trx-MYOT (Trx-MYOT-NEECK), possessing mutations resembling those in the ACTN2-NEECK (see Fig 4E). Lightning bolt depicts position of mutations. **(D)** Effects of myotilin on α-actinin–F-actin interaction. ACTN2-NEECK was added to F-actin at increasing concentrations before (ACTN2-NEECK before tropomyosin) incubation of F-actin with the fixed amount of tropomyosin. Mean values (± SEM) of 3 independent experiments are shown. The mean binding of tropomyosin to F-actin at all concentrations of ACTN2-NEECK was significantly different from the means in its absence. Significance was assessed using 1-way ANOVA with Holm–Sidak test, $p < 0.05$ for 0 vs. 1.2 μM ACTN2-NEECK, $p < 0.01$ for 0 vs. 2.4 μM ACTN2-NEECK, and $p < 0.001$ for 0 vs. all other concentrations of ACTN2-NEECK. **(E)** Sequence alignment of proteins and their residues involved in binding to α-actinin EF34. Residues of the CaM-binding motif 1-4-5-8 are shown in bold and boxed in green. Residues mutated in α-actinin-2 NEECK and myotilin NEECK are shown in red. **(F)** Binding of myotilin to α-actinin. ACTN2-WT, ACTN2-NEECK, for details see (C and E), and ACTN2-EF14 were subjected at increasing concentrations to pull-down assay with (bound) or without (control) Trx-MYOT. The uncropped gels can be found in S1 Data. **(G and H)** Results of ITC experiments quantifying the interaction between (G), Trx-MYOT, or (H) its mutant (Trx-MYOT-NEECK), for details, see (C and E) and ACTN2-EF14. n.d., not determined. Data points that were used to create graphs are reported in S2 Data. See also S4 Fig. The 3D model presented in this panel is available in the following link: https://skfb.ly/6YGQt. ACTN2-EF14, α-actinin-2 EF14; ACTN2-WT, wild-type α-actinin-2; Trx, thioredoxin; WT, wild-type.

constitutively open mutant ACTN2-NEECK, or CAMD encoding all 4 EF-hands of α-actinin-2 (ACTN2-EF14), with Trx-MYOT (Fig 4C and 4E). We observed myotilin binding to ACTN2-NEECK and ACTN2-EF14, but not to ACTN2-WT (Fig 4F), indicating that opening of α-actinin-2 is necessary for its interaction with myotilin, as was also revealed for titin [8].

To validate the binding site between α-actinin-2 and myotilin (Fig 4C and 4E), we next performed isothermal titration calorimetry (ITC) experiments with ACTN2-EF14, Trx-MYOT, and its mutant variant (Trx-MYOT-NEECK), where mutations resembling those in ACTN2-NEECK were introduced in the binding motif to disrupt the interaction. Indeed, the affinity of Trx-MYOT to α-actinin-2 was found to be in μM range ($K_d = 4.2 ± 1.2$ μM, Fig 4G), whereas no binding of Trx-MYOT-NEECK to ACTN2-EF14 was observed (Fig 4H).

Altogether, our data reveal that myotilin, which contains the conserved α-actinin-2 binding motif, binds to the open confirmation of α-actinin-2 via pseudoligand regulatory mechanism.

## Interaction of myotilin with F-actin is not regulated by PI(4,5)P$_2$

The Ig3 domain of palladin is the minimum fragment necessary for binding to F-actin [19,29]. Furthermore, Ig3 interacts with the head group of PI(4,5)P$_2$ with a moderate affinity ($K_d = 17$ μM) leading to a decreased F-actin cross-linking and polymerization activity [48]. Myotilin binding to α-actinin-2 follows the same PI(4,5)P$_2$ mechanism as titin, while its F-actin binding is similar to that of palladin. Therefore, we investigated whether myotilin binds PI(4,5)P$_2$ as well, using a liposome co-sedimentation assay [48]. The amount of protein bound to liposomes was examined by using a constant concentration of Ig1Ig2$^{250–444}$, while varying the PI(4,5)P$_2$ concentration (0% to 20%) in 1-palmitoyl-2-oleoyl-sn-glycero-3-phosphocholine (POPC) vesicles. As positive controls, double C2-like domain-containing protein beta (Doc2b), a known sensor of PI(4,5)P$_2$ [49], and the Ig3 domain of palladin were used (S4D and S4E Fig). Both

proteins, but not myotilin Ig1Ig2$^{250-444}$, co-pelleted with PI(4,5)P$_2$-POPC vesicles in a PI(4,5)P$_2$ concentration-dependent manner (S4D and S4E Fig), indicating that myotilin does not interact with PI(4,5)P$_2$. This observation was further corroborated by structure-based sequence analysis, which showed that residues important for binding of PI(4,5)P$_2$ to palladin [48] are not present in the Ig2 domain of myotilin (S4F Fig). Therefore, the myotilin interaction with F-actin is most likely not regulated by PI(4,5)P$_2$.

## Binding to F-actin affects myotilin mobility and dynamics in cells

To assess whether actin-binding affects the cellular dynamics of myotilin, we expressed N-terminal EGFP fusion of myotilin, or its mutants that showed impaired F-actin binding activity in vitro (K354A, K359A, K354/359A, and K354/358/359A), in differentiating C2C12 mouse myotubes. After 7 days of differentiation, all of the tested mutants and the wild-type myotilin showed a similar distribution and localized to Z-discs (Fig 5A, panel "pre-bleaching"), suggesting that similarly to EGFP, N-terminal Trx-tag used for in vitro studies does not perturb myotilin function. Fluorescence recovery after photobleaching (FRAP) was performed to compare the mobility and dynamics of the mutant variants with wild-type myotilin (Fig 5A). Wild-type myotilin showed rapid dynamics with 80% fluorescence recovery after 300 s (median half-time $t_{1/2}$ of 72.4 s) (Fig 5B), which is in excellent agreement with previously published reports [50,51]. While the single mutant K359A only showed slight, nonsignificant changes ($t_{1/2}$ 64.1 s), the K354A showed a significantly faster recovery with $t_{1/2}$ of 48.2 s (Fig 5B and 5C). Protein dynamics further increased proportionally to the number of mutations in double ($t_{1/2}$ 35.5 s) and triple ($t_{1/2}$ 32.3 s) mutants (Fig 5B and 5C). The differences in recovery times are evident by comparing the fluorescence signals in Fig 5A: The wild-type and K359A myotilin hardly showed any recovery after 5 or even 20 s, indicating relatively slow dissociation and high affinity, but a clear signal can be seen for the other 3 mutants with reduced actin binding strength. At the same time, the mobile fraction of the mutants increased correspondingly (Fig 5D).

These results show that mutant myotilin variants with weaker binding to F-actin in vitro have significantly altered dynamics in the sarcomeric Z-disc, indicating that binding to F-actin stabilizes myotilin in the sarcomere.

## Discussion

Biochemical and structural analysis of myotilin and its interactions presented in this study provide a basis for the mechanistic understanding of its structural role in the sarcomeric Z-disc as a protein–protein interaction platform and inferring its role in orchestration of Z-disc assembly and regulation of sarcomere biogenesis as a distinctive novel function of myotilin.

Our studies hint on Ig2 as the central F-actin interaction region, predominantly via electrostatic interactions. In addition, they indicate that tandem Ig1Ig2 is the minimal functional F-actin–binding module, which the N-terminal and carboxyl-terminal flanking regions further enhance binding affinity and thus contribute to complex stability. This is supported by a body of evidence: (i) positively charged patches are on the same surface on both Ig domains; (ii) XL-MS maps specific cross-links between F-actin and Ig1, Ig2 as well as the flanking regions; (iii) the flanking regions enhance binding to F-actin; (iv) NMR substantiated the interaction sites to map to the same face on both Ig domains; and (v) binding of tandem Ig1Ig2 to F-actin stabilizes F-actin and prevents its depolymerization. The latter may be explained by binding of each Ig domain to 1 actin subunit, allowing the tandem Ig1Ig2 to bridge 2 actin subunits and thus act as a vertical cross-linker. The importance of tandem Ig domains for interaction with F-actin was also observed for palladin where Ig3Ig4 tandem displayed enhanced binding in comparison to Ig3 alone [19]. In addition, our FRAP experiments highlight the importance of

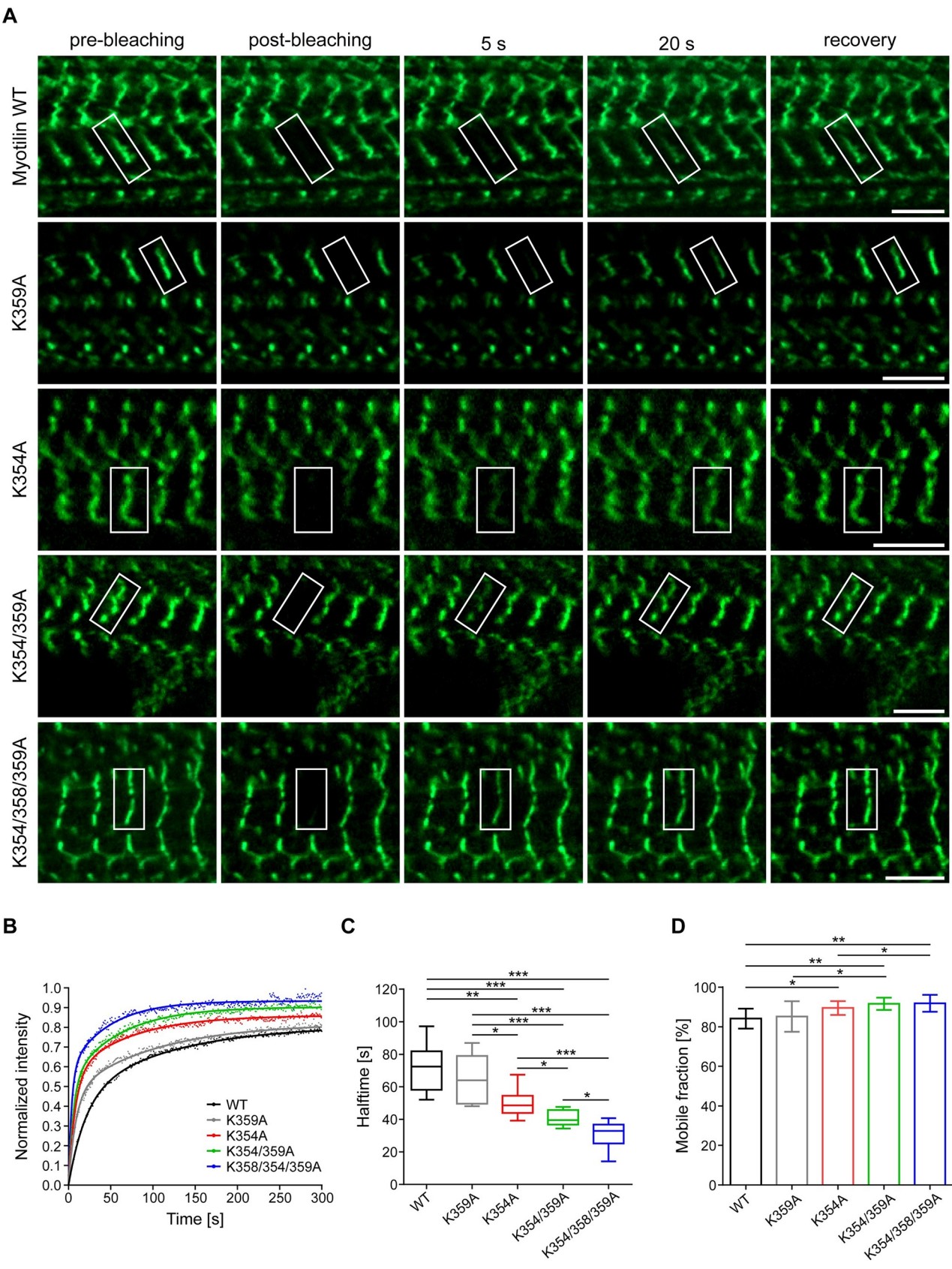

**Fig 5. Binding of myotilin with F-actin affects its mobility and dynamics. (A)** FRAP analysis of the dynamics of myotilin and its mutant variants in Z-discs. Localization of EGFP fusion proteins of WT myotilin and its variants with the indicated mutations in Ig2 before bleaching (pre-bleaching), directly after bleaching (post-bleaching), and after the indicated time points after bleaching. White rectangles show ROIs, used for bleaching during FRAP experiments. Scale bars, 5 μm. **(B)** Representative FRAP recovery curves of the myotilin variants. **(C)** Half-times of myotilin mutants compared to WT myotilin. Quantification of data from FRAP studies shown in (A and B). Statistical data are depicted in box and whisker plots. Calculated median half-times are shown as a line surrounded by a box, representing the interquartile range comprising the median ± 25% of the data. Whiskers extend at most 2 standard deviations from the median. **(D)** Mobile fractions of mutants compared to WT myotilin. Graph reports mean ± SEM, $n = 8$ myotubes for each variant. A total of 1–3 Z-discs were bleached per myotube. Significance was assessed using 2-tailed Student $t$ test, $^{*}p < 0.1$, $^{**}p < 0.05$, and $^{***}p < 0.01$. Data points that were used to create graphs are reported in S2 Data. FRAP, fluorescence recovery after photobleaching; ROI, region of interest; WT, wild-type.

the positively charged patch in Ig2 for binding. Proteins with charge-neutralizing mutations in Ig2 displayed increased mobile fractions and shorter fluorescence recovery times in differentiated C2C12 cells, indicating reduced binding affinity.

von Nandelstadh and colleagues (2005) previously suggested Ig1Ig2 tandem to be important for optimal interaction with F-actin and showed that the size of myotilin constructs correlates with binding efficiency where longer fragments bind to F-actin stronger [18]. However, a detailed interaction map and molecular mechanism of the binding were not known. Based on our findings, we generated the first molecular model of F-actin decorated by tandem Ig domains, which together with our quantitative binding assays allowed us to propose a mechanism for myotilin:F-actin assembly. In the unbound state, flexibility and preferential extended conformation give myotilin a high level of structural plasticity, which is an important component of its binding partner recognition mechanism. In the initial step of the interaction with F-actin, the positively charged region on myotilin Ig2 as the predominant binder attaches to the negatively charged subdomain 1 of actin. The subsequent Ig1:F-actin interaction enables a transition to a completely extended myotilin conformation and consequently, stabilization of both Ig domains plus the flanking regions on F-actin.

Although palladin and myotilin are closely related, and the interaction with F-actin is in both cases mediated by a conserved mechanism involving basic charged clusters in the Ig domains, they seem to differ fundamentally in the precise geometry and stoichiometry of the complexes formed. Palladin binds actin via 2 basic clusters situated on 2 sides of a single Ig domain [29], allowing for cross-linking 2 actin filaments with a single palladin molecule. In contrast, a single myotilin molecule harbors only 1 high-affinity binding site. Therefore, a dimerization is required to enable F-actin cross-linking. Furthermore, the contribution of Ig domain flanking regions to F-actin binding may allow for regulation by posttranslational modifications (PTMs) [4,5].

Since the interaction of most actin-binding proteins with F-actin may either be regulated by certain compounds or modulated by other proteins, it is essential to contemplate the precise situation in the cell. For myotilin, it is therefore important to investigate potential roles of PIPs and known binding partners like α-actinin-2. α-Actinin-2 recognizes the 1-4-5-8 motif of the cognate α-actinin-2 "neck" and/or titin Zr-7 by the pseudoligand mechanism regulated by PI(4,5)P$_2$. Our studies suggest that myotilin binds to α-actinin-2 CAMD employing the same regulatory mechanism, setting the molecular basis of interaction for the entire palladin family. In addition, our binding data suggest that titin can displace myotilin from α-actinin-2, as titin Zr-7 interacts with α-actinin-2 with a stronger affinity (0.48 ± 0.13 μM) [8] than myotilin.

Furthermore, binding of PI(4,5)P$_2$ to α-actinin-2 down-regulates its F-actin binding activity, resulting in decreased F-actin cross-linking [52]. In palladin, PI(4,5)P$_2$ inhibits palladin-mediated F-actin polymerization and cross-linking [48]. In contrast, we showed that myotilin Ig1Ig2 does not bind PI(4,5)P$_2$; however, interactions with other parts cannot be completely excluded, suggesting that the myotilin–F-actin interaction is not regulated by PI(4,5)P$_2$, as

opposed to α-actinin-2 and palladin. This implies a possible mechanism for stabilization of α-actinin-2 on F-actin upon binding to myotilin: PI(4,5)P$_2$ binding induces α-actinin-2 open conformation enabling it to interact with myotilin, while its F-actin binding is simultaneously reduced due to PI(4,5)P$_2$. Concurrent binding of myotilin to α-actinin-2 and F-actin can thus help to stabilize α-actinin-2 on F-actin and may compensate for the inhibitory effect of PI(4,5)P$_2$.

Our SAXS data revealed full-length myotilin as a dynamic ensemble of multiple conformations with local structurally compact clusters, as previously predicted based on sequence analysis [23]. Further local induction of structure might be promoted upon binding, in particular to α-actinin-2, similar to titin and palladin where the interaction with CAMD of α-actinin-2 induces folding of the binding motif into an α-helical structure [8,45,47].

Our studies of myotilin dimerization in solution revealed that the regions flanking Ig1Ig2 domains enhance dimer stability. In comparison to the previous studies addressing myotilin dimerization [10,11,18], we show that myotilin dimerizes in a concentration-dependent manner and that only concentrations higher than 1.0 mM showed complete dimer population, implying a weak association. This suggests mechanisms driving or stabilizing dimer formation in vivo: high local concentrations and avidity effects mediated by concomitant binding to interaction partners such as α-actinin, filamin C, and of course F-actin that tether myotilin to the Z-disc, and/or PTMs.

All known pathogenic myotilin variants result in a single residue substitution in the protein, and all, except R405K, map to the N-terminus, within or close to the serine-rich region (SRR) and the hydrophobic residues-rich region (HRR) representing its "mutational hotspot" (Fig 1B). These mutations are predominantly substitutions from a polar/charged to a hydrophobic residue. Consequently, in concert with the slower degradation and changed turnover as already indicated [23,24], a newly exposed hydrophobic cluster could promote aggregation and play an important role in pathological mechanisms. The R405K mutation in Ig2 was suggested to be responsible for defective homodimerization of myotilin and decreased interaction with α-actinin-2 [27]. However, our results revealed that R405K mutation does not impair myotilin homodimerization in vitro (S1G Fig), nor does it affect the binding of myotilin to F-actin (Fig 2A), which was expected since a positively charged residue is retained at this position. In addition, we predict that this mutation will not have an impact on binding of myotilin to α-actinin-2, since it does not reside in the α-actinin-2 binding site nor its proximity. Thus, further in vitro and in vivo experiments are needed to understand the molecular mechanism underlying the pathophysiology caused by the R405K mutation.

An intriguing yet unresolved question is: How are the multitude of F-actin–binding proteins sorted along thin myofilaments and Z-discs? Strikingly, tropomyosin is distributed all along thin filaments except the Z-disc region [7]. In non-muscle cells, tropomyosin competes with α-actinin-1 and α-actinin-4, for F-actin binding, resulting in mutually exclusive localizations along stress fibers [36, 37]. We reveal that striated muscle-specific α-actinin-2 also competes with muscle type tropomyosin for binding to F-actin, and thus extend the competition mechanism to all isoforms of α-actinin. Most importantly, we found that myotilin also competes with tropomyosin for F-actin binding and stabilizes F-actin against depolymerization, suggesting that it may be involved in excluding tropomyosin from the Z-disc. Thus, binding of myotilin may create functionally distinct regions on actin filaments, thereby modulating the interactions of other proteins with F-actin. However, myotilin alone does not explain the absence of tropomyosin from the Z-disc. Instead, a concerted action of several Z-disc F-actin–binding proteins (e.g., filamin C, α-actinin-2, ZASP, myotilin, etc.) is required, which is supported by the finding that the tropomyosin binding site on F-actin partly overlaps with those for filamin C, α-actinin-2, and myotilin (S4C Fig). Sorting would be facilitated by local protein

complex formation, resulting in increased F-actin affinity, based on avidity effects. In this context, it is noteworthy that even a dramatic reduction of the binding strength of myotilin to F-actin, as shown by our FRAP experiments, does not impair its Z-disc localization (Fig 5), supporting the view that the Z-disc is an intricate network of interacting multidomain proteins, which may tolerate reduction or even loss of individual interactions without attenuating the entire structure. This robustness rooted in redundancy may also explain that the loss of a single protein by gene knockout may be compensated by other proteins [53]. Accordingly, the lack of a phenotype in myotilin knockout mice may be compensated by myopalladin and/or palladin, which interact with EF34 hands of α-actinin-2 with similar affinity as myotilin [47].

This raises the question about the role of myotilin in Z-disc biogenesis. According to current models, in early myofibrillogenesis, premyofibrils are composed of minisarcomeres containing a subset of sarcomeric proteins in α-actinin-2-enriched Z-bodies, whereas the attached thin filaments are associated with tropomyosin and troponin [54]. A recent study showed that in the cell, tropomyosins are present at sufficient levels to saturate all actin filaments [55]. Therefore, tropomyosin needs to be actively inhibited from binding to specific regions on actin filaments to allow the formation of tropomyosin-free actin filaments [55] necessary for the development of Z-discs. Myotilin is expressed at relatively late stages of differentiation, when α-actinin-2, the Z-disc portion of titin, myopodin/SYNPO2, and filamin C already co-localize at Z-bodies [11,56,57]. Myotilin entry into this still relatively loose protein assembly occurs at the time when mechanical strain increases due to incipient contractile activity. Its simultaneous binding to F-actin and α-actinin, as well as the PIP$_2$-dependent Zr7–α-actinin-2 interaction and the simultaneous displacement of tropomyosin, make myotilin an ideal co-organizer of Z-disc proteins (Fig 6). This timing seems essential since the longer α-actinin molecules organize the more widely spaced F-actin networks required for myofibrils, whereas myotilin alone would form too tightly packed filament bundles. Further studies are required to determine the precise sequence of events regarding the assembly of components in maturing Z-discs.

## Materials and methods

### Plasmids and DNA constructs

Different DNA fragments, encoding human myotilin, or α-actinin-2 were subcloned into different vectors as specified in S3 Table. For preparation of myotilin mutants (S3 Table), mutagenesis was performed with the QuikChange II Site-Directed Mutagenesis Kit (Agilent Technologies, United States of America). Trx-MYOT, MYOT$^{WT}$, and Ig1Ig2$^{250-444}$ constructs were used as templates in the PCR reactions. Correct assembly of the prepared constructs and mutagenesis efficiency was verified by DNA sequencing (GATC Biotech, Germany).

### Protein expression and purification

All recombinant myotilin and α-actinin-2 constructs were expressed in *Escherichia coli* (see S3 Table). Full-length myotilin (Trx-MYOT) with N-terminal Trx tag and its mutant variant Trx-MYOT-NEECK were expressed in B834 (DE3) grown at 37˚C in LB media to an OD$_{600}$ of approximately 1.0. Protein expression was induced by addition of IPTG to a final concentration of 0.5 mM, and cells were grown for additional 4 to 5 h at 37˚C. Proteins were further purified by affinity chromatography using HisTrap FF columns (Cytiva, USA) followed by size exclusion chromatography (SEC) on a HiLoad 26/600 Superdex 200 column (Cytiva) equilibrated with 20 mM Tris, 400 mM NaCl, 250 mM arginine, 2 mM β-mercaptoethanol, pH 7.5. Ig1Ig2$^{185-498}$ was expressed in C41 (DE3) strain following the protocol described in [8] and purified by affinity chromatography using HisTrap FF columns. Subsequently, the Trx-tag was

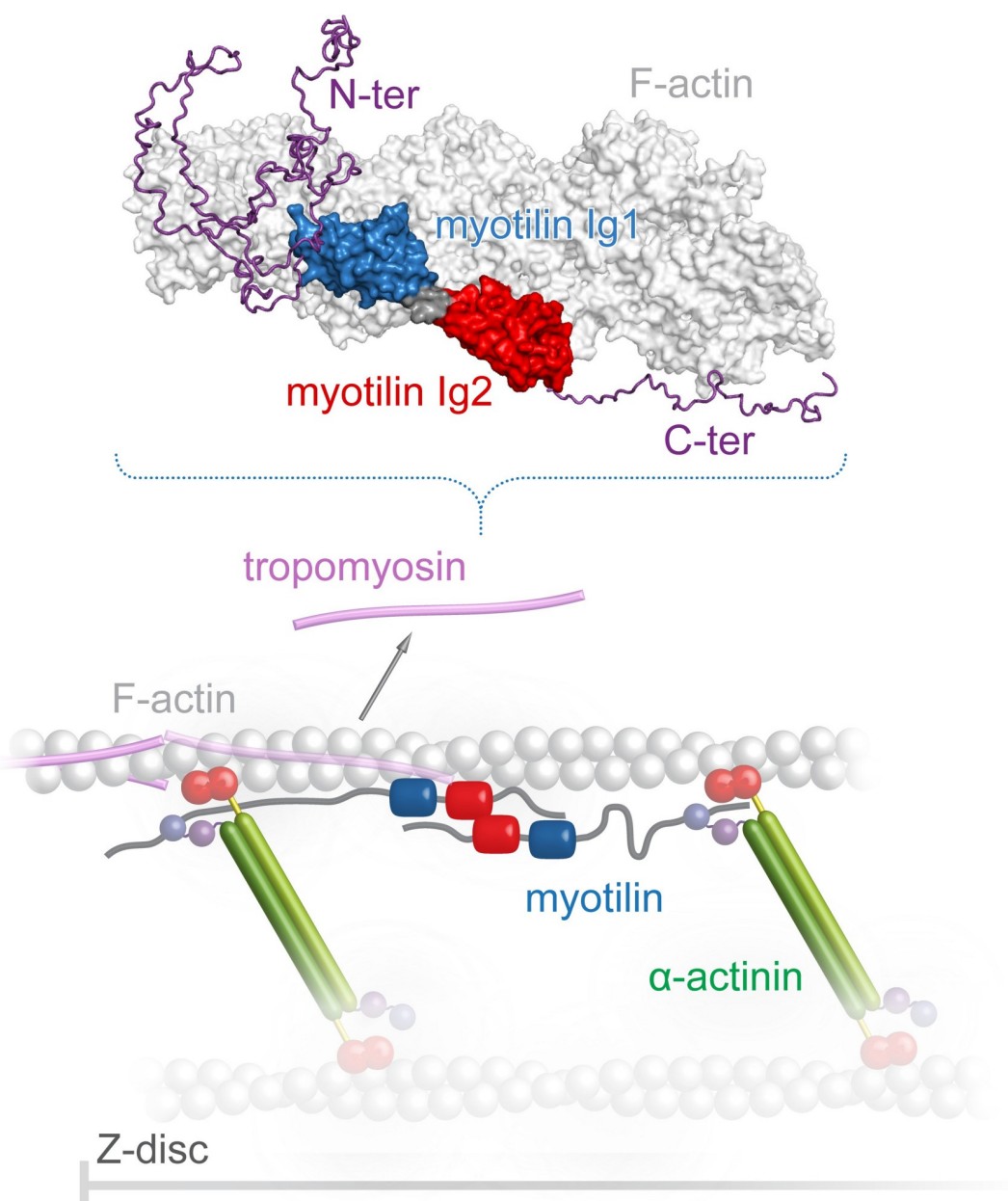

**Fig 6. Model illustrating the role of myotilin in the organization of sarcomeric Z-disc assembly.** Myotilin is recruited to the Z-disc by binding to F-actin, α-actinin-2, and/or other Z-disc components e.g., filamin C, ZASP. As a part of the recruitment process, myotilin can further stabilize (in its propose antiparallel dimeric form) [11] 2 α-actinin-2 dimers on F-actin through interaction of its N-terminal region with CAMD of α-actinin-2. Consequently, together with α-actinin-2 and/or other Z-disc components, myotilin can help to displace tropomyosin from the Z-disc. For simplicity, only some of the Z-disc components are shown. α-Actinin-2 dimers are schematically shown in an open conformation bound to myotilin, with ABD highlighted as red spheres and EF12 and EF34 of CAMD as violet and purple spheres, respectively. Myotilin and tropomyosin are shown in color code as used in Figs 1B and 4A. The 3D model presented in this panel is available in the following link: https://skfb.ly/6YGQF. ABD, actin binding domain; CAMD, calmodulin-like domain.

cleaved overnight with HRV-3C protease added at a mass ratio protease:protein of 1:1,000, and the Ig1Ig2$^{185-498}$ was further purified by StrepTrap HP columns (Cytiva), followed by SEC on a HiLoad 26/600 Superdex 75 column (Cytiva) equilibrated with 20 mM Tris, 150 mM NaCl, 5% glycerol, pH 7.5.

During purification, the Ig1Ig2$^{185-498*}$ readily degrades from its carboxyl terminus and forms Ig1Ig2$^{185-454}$ as determined by intact protein mass spectrometry analysis of the final product. Therefore, to prepare Ig1Ig2$^{185-454}$, the Ig1Ig2$^{185-498*}$ construct was expressed using the procedures adapted from Drmota Prebil and colleagues (2016) [58]. After expression, cells were lysed by sonication, and insoluble material was removed by centrifugation. The lysate was loaded onto a GSTrap FF column (Cytiva) equilibrated with 10 mM sodium phosphate, 140 mM NaCl, 2.7 mM KCl, pH 7.4 (PBS). After washing with the same buffer, the bound proteins were eluted by a linear gradient of 50 mM Tris, 10 mM reduced glutathione, pH 8.0. His$_6$-GST-tag was removed by incubation with His$_6$-tagged TEV protease (added at a mass ratio protease:protein of 1:100) during overnight dialysis against the dialysis buffer (50 mM Tris, 150 mM NaCl, 1 mM DTT, pH 7.4) at 4˚C. The His$_6$-GST-tag-free Ig1Ig2$^{185-454}$ was recovered as a flow-through after applying the cleavage mixture onto Ni$^{2+}$-loaded HiTrap IMAC FF (Cytiva) and GSTrap FF columns to remove TEV protease, uncleaved proteins, and the cleaved-off His$_6$-GST moiety. Final purification step was SEC on a Superdex 75 10/300 column (Cytiva) equilibrated with 20 mM HEPES, 150 mM NaCl, 5% glycerol, 1 mM DTT, pH 7.4.

During expression, Ig1Ig2$^{250-498}$ degrades to Ig1Ig2$^{250-466}$ as verified by intact protein mass spectrometry analysis. Thus, depending on the expression/purification strategy either Ig1Ig2$^{250-498}$ or Ig1Ig2$^{250-466}$ can be prepared from cells expressing Ig1Ig2$^{250-498}$. To obtain Ig1Ig2$^{250-466}$, the expression and purification procedure is basically the same as in [23] with the following differences: (1) for lysis buffer, 20 mM HEPES, pH 7.4, 150 mM NaCl, 5% glycerol, 10 mM imidazole was used, the lysate was applied to HisTrap column, and for elution, the same buffer supplemented with 180 mM imidazole was used; (2) cleavage with HRV-3C protease was done at protease:protein mass ratio 1:50 using 20 mM HEPES, 150 mM NaCl, 5% glycerol, pH 7.4; and (3) the final SEC step was done using a HiLoad 16/600 Superdex 75 column (Cytiva) equilibrated with 20 mM HEPES, 150 mM NaCl, 5% glycerol, pH 7.2. The same procedure was used to prepare Ig1Ig2$^{220-452}$ and Ig1Ig2$^{220-452\ R405K}$. To obtain Ig1Ig2$^{250-498}$, the procedure was the same as for Ig1Ig2$^{250-466}$; however, protein expression was done in B834 (DE3) strain at 37˚C for 3 h after induction with 0.5 mM IPTG. In addition, before final SEC step, Ig1Ig2$^{250-498}$ was purified by cation exchange chromatography using Resource S column (Cytiva) in 10 mM HEPES, 2 M urea, 5% glycerol, and gradient of 0–1 M NaCl, pH 7.0.

The Ig1Ig2$^{250-444}$ and all its mutant variants (see S3 Table) were purified as described elsewhere [23]. The Ig1$^{250-344}$ and Ig2$^{349-459}$ constructs were purified using the procedures adapted from Drmota Prebil and colleagues (2016) [58]; the only modification was in the final SEC step where the buffer was replaced by 20 mM HEPES, 150 mM, NaCl, 5% glycerol, 1 mM DTT, pH 7.4.

Full-length α-actinin-2 constructs (ACTN2-WT and ACTN2-NEECK) were prepared as described previously [8]. ACTN2-EF14 was expressed as described in [8] and purified by affinity chromatography using HisTrap FF columns. Subsequently, the His$_6$-tag was removed by incubation with HRV-3C protease added at a mass ratio protease:protein of 1:50 during overnight dialysis against PBS, 5% glycerol, 1 mM EDTA, 1 mM DTT at 4˚C. The His$_6$-tag-free ACTN2-EF14 was recovered as a flow-through after applying the cleavage mixture onto HisTrap FF column to remove the HRV-3C protease and uncleaved proteins. Final purification step was SEC on a HiLoad 26/600 Superdex 75 column equilibrated with PBS, 5% glycerol.

Palladin Ig3 was kindly provided by Moriah R. Beck (Wichita State University, Wichita, USA) and prepared/purified as described in [48]. Plasmid encoding constitutively monomeric mutant of actin (DVD-actin) was kindly provided by Michael K. Rosen (University of Texas Southwestern Medical Center, Dallas, USA), and protein was prepared as described in [31].

Doc2b [49] and tropomyosin [34] were kindly provided by Sascha Martens (Max Perutz Labs, University of Vienna, Vienna, Austria) and Stefan Raunser (Max Planck Institute of Molecular Physiology, Dortmund, Germany), respectively. Actin was prepared from rabbit skeletal muscle [59] and pyrene-labeled following [60].

## SAXS data collection and analysis

SEC-SAXS data collection of Trx-MYOT in 20 mM Tris, 400 mM NaCl, 250 mM arginine, 5% glycerol, pH 7.5 was conducted at the EMBL P12 beamline of the storage ring PETRA III (DESY, Hamburg, Germany) [61] using an incident beam size of $200 \times 110$ μm$^2$ (full width at half maximum, FWHM) in a 1.7-mm quartz capillary held under vacuum. For this, the setup as described in [62] was employed. Here, an integrated micro-splitter (P-451, Upchurch Scientific, USA) allows the eluent of a chromatography column (Superdex 200 increase 10/300 (Cytiva)) to flow equally through the SAXS capillary and a modular triple detector array (TDA, Viscotek model TDA 305, Malvern Panalytical, United Kingdom) that extracts molecular mass estimates of SEC-separated components by correlating refractive index (RI) and/or UV-vis concentrations with RALLS using the integrated Omnisec software (Malvern Panalytical). In parallel, 3,000 individual SAXS frames were collected with 1-s exposure that were used for subsequent SAXS analysis. The chromatography was conducted at 0.5 ml/min. SAXS data reduction to produce the final scattering profile of monomeric Trx-MYOT was performed using standard methods. Briefly, 2D-to-1D radial averaging was performed using the SAS-FLOW pipeline [63]. Aided by the integrated prediction algorithms in CHROMIXS, the optimal frames within the elution peak and the buffer regions were selected [64]. Subsequently, single buffer frames were subtracted from sample frames one by one, scaled, and averaged to produce the final subtracted curve, which was further processed with various programs from the ATSAS software package [63]. Details on data collection are shown in S1 Table.

SAXS data for purified myotilin constructs Ig1Ig2$^{250-444}$ and Ig1Ig2$^{250-498}$ were collected at ESRF beamline BM29 BioSAXS (Grenoble, France) equipped with the Pilatus 1M detector. Samples were measured at concentrations of up to 14.12 (Ig1Ig2$^{250-444}$) or 42.8 mg/ml (Ig1Ig2$^{250-498}$) in 20 mM HEPES, 150 mM, NaCl, 5% glycerol, 1 mM DTT, pH 7.4, in 2 independent measurements. One SAXS dataset for the purified myotilin construct Ig1Ig2$^{220-452}$ was collected at the EMBL X33 beamline of the storage ring Doris (DESY) equipped with the Pilatus 1M-W detector at concentrations in the range from 1 to 52 mg/ml in 20 mM MES, 200 mM NaCl, 3% glycerol, pH 6.0. The other SAXS dataset for the myotilin construct Ig1Ig2$^{220-452}$ and the Ig1Ig2$^{220-452\ R405K}$ mutant was collected at EMBL P12 beamline of the storage ring Petra III (DESY) equipped with the Pilatus 6M detector, in the concentration range 2.1 to 36.6 mg/ml (Ig1Ig2$^{220-452}$) and 2.3 to 45.5 mg/ml (Ig1Ig2$^{220-452\ R405K}$) in 20 mM HEPES, 150 mM NaCl, 5% glycerol, 1 mM DTT, pH 7.4. The momentum of transfer $s$ is defined as $s = 4\pi sinq/\lambda$. Details on data collection are shown in S1 Table. For all constructs, background scattering was subtracted, data reduced, normalized according to the measured concentration, and extrapolated to infinite dilution using the 2 lowest measured concentrations using PRIMUS [65] module of the ATSAS software package [63]. Forward scattering (I$_0$) and radius of gyration (R$_g$) were obtained by fitting the linear Guinier region of the data. Pair distribution function $P(r)$ with the corresponding maximum particle size parameter (D$_{max}$) was determined using GNOM program [66].

For reconstruction of a theoretical molecular envelope, ab initio modeling was performed 20 times using the program DAMMIF [67], where scattering from the calculated envelopes was fitted against the experimental scattering and evaluated by the χ values. The most typical envelope was selected by comparing the normalized spatial discrepancy (NSD) values between

pairs of envelopes and later averaged by DAMAVER set of programs [68]. In rigid body modeling, high-resolution structures of myotilin Ig1 (PDB: 2KDG) and Ig2 (PDB: 2KKQ) were used to fit SAXS scattering data using program CORAL [69]. Linker residues were designated as dummy atoms. Ab initio calculated envelope was superposed to the rigid body model using SUPCOMB program [70].

Flexibility analysis for Trx-MYOT was performed using EOM 2.1 program [26] with enforced P1 symmetry, while the linker between Ig domains was again designated as dummy atoms. To account for the available information about the structure, the ensembles were generated as follows: During the generation, the potentially flexible termini/domain were allowed to have randomized conformations. Conformations consistent with scattering data were selected from the pool of 50,000 models using a genetic algorithm. The flexibility analysis was independently repeated 3 times; all runs gave comparable results. One should, however, note that for flexible macromolecules exploring a range of conformations in solution, shape restoration returns average over the ensemble at a low resolution of the model in Fig 1D of 56 ± 4 Å. All structure figures were prepared using PyMOL (The PyMOL Molecular Graphics System, version 2.3, Schrödinger). The SAXS data were deposited at SASBDB under the accession codes SASDFZ7, SASDF28, SASDF38, SASDF48, SASDJH8, and SASDJJ8.

## Cross-linking and mass spectrometry analyses

For cross-linking experiments with F-actin, actin was let to polymerize in PBS for 30 min at room temperature before adding 4-(4,6-dimethoxy-1,3,5-triazin-2-yl)-4-methylmorpholiniumchloride (DMTMM, Merck, USA) cross-linker. After incubation of F-actin with DMTMM for 5 min at room temperature, Ig1Ig2$^{250-444}$, or Ig1Ig2$^{185-454}$, or Ig1Ig2$^{185-498}$ in PBS was added at concentrations indicated in the S3 Fig, and proteins were cross-linked for additional 40 min at room temperature. While for cross-linking of Ig1Ig2$^{250-444}$, increasing concentrations of DMTMM were used (S3A Fig), fixed concentration of DMTMM (2 mM) was used to cross-link Ig1Ig2$^{185-454}$ and Ig1Ig2$^{185-498}$ with F-actin (S3B and S3C Fig). Following separation of cross-linking samples by SDS-PAGE and staining of proteins with colloidal Coomassie Blue G-250, protein bands were excised, subjected to in-gel digestion using trypsin, and analyzed by HPLC–ESI–MS/MS using an UltiMate 3000 RSLCnano system coupled to a Q Exactive Plus (both Thermo Fisher Scientific, USA) mass spectrometer as described [71]. For identification of cross-linked peptides, the software pLink [72] was used, version 1.22 for Ig1Ig2$^{250-444}$ and Ig1Ig2$^{185-454}$ and version 2.0 for Ig1Ig2$^{185-498}$. For the former, MS raw data files were first converted to the Mascot generic format (mgf) using msConvert from ProteoWizard, release 3.0.9740 [73]. Searches were performed against the forward and reversed amino acid sequences of the recombinant proteins with parameters and filtering criteria as described previously [74]. In brief, cross-linked peptide matches were filtered at a false discovery rate (FDR) of 5%, and a minimum E-value of $5 \times 10^{-2}$ rejecting adjacent tryptic peptides of the same protein.

## Actin co-sedimentation assays

Actin co-sedimentation assays were performed as published in [75], with some modifications. As various myotilin fragments displayed different solubility mostly depending on the pH of the buffer used within the assay, co-sedimentation assays were performed at 2 different conditions, B1 and B2. In the B1 conditions, rabbit skeletal muscle actin in G-buffer (2 mM HEPES, pH 8.0, 0.2 mM CaCl$_2$, 0.2 mM ATP, 2 mM β-mercaptoethanol) was polymerized with the addition of 1/10 volume of 10× F-buffer (100 mM HEPES, pH 7.4, 500 mM KCl, 20 mM MgCl$_2$) and incubated for 30 min at room temperature. Purified myotilin constructs Ig1$^{250-}$

$^{344}$, Ig2$^{349-459}$, and Ig1Ig2$^{250-444}$ and its mutant variants were dialyzed overnight against the dialysis buffer (20 mM HEPES, pH 7.4, 100 mM NaCl, 5% glycerol, 1 mM DTT). For apparent dissociation constant determination, fixed concentration (16 μM) of myotilin constructs in 20 μL of dialysis buffer was mixed with pre-polymerized F-actin (0–120 μM) in 20 μl of 1× F-buffer. For co-sedimentation with the increasing salt concentrations, pre-polymerized actin (6 μM) in 60 μl of 1× F-buffer was added to the myotilin construct in 60 μl of 1× F-buffer supplemented with or without KCl to achieve a final concentration of 50, 100, 150, 250, and 300 mM. For determination of binding of mutant myotilin variants to F-actin, pre-polymerized actin (8 μM) in 60 μl of 1× F-buffer were added to the mutant myotilin variants (8 μM) in 60 μl of dialysis buffer. In the B2 conditions, the G-buffer was 2 mM PIPES, pH 6.8, 0.2 mM CaCl$_2$, 0.2 mM ATP, 5 mM β-mercaptoethanol, 10× F-buffer consisted of 100 mM PIPES, 1 M NaCl, 10 mM MgCl$_2$, 10 mM EGTA, pH 6.8, and myotilin fragments Trx-MYOT, Ig1Ig2$^{250-444}$, Ig1Ig2$^{250-498}$, Ig1Ig2$^{185-454}$, and Ig1Ig2$^{185-498}$ were dialyzed overnight against the 10 mM PIPES, 100 mM NaCl, 1 mM MgCl$_2$, 1 mM EGTA, 5% glycerol, pH 6.8. For determination of apparent dissociation constant(s) at B2 conditions, fixed concentration (6 μM) of myotilin fragment in 20 μl of dialysis buffer was mixed with pre-polymerized F-actin (0 to 60 μM) in 20 μl of 1× F-buffer. Co-sedimentation assays with tropomyosin and Trx-MYOT or ACTN2--NEECK mutant of α-actinin-2 were performed at B2 conditions as well. Here, Trx-MYOT (at final concentration 0 to 4.8 μM) or ACTN2-NEECK (0 to 6.0 μM) were mixed with pre-polymerized actin (6 μM) 30 min before the addition of tropomyosin (2 μM), or tropomyosin (2 μM) was mixed with pre-polymerized actin (6 μM) 30 min before addition of Trx-MYOT (at final concentration 0 to 4.8 μM). Reaction mixtures in all co-sedimentation assays were incubated at room temperature for 30 min prior to the ultracentrifugation (125,000 × g, 30 min, 22°C). Pellets and supernatants were separated and equal volumes analyzed by SDS-PAGE. Gels were scanned and quantified by densitometry with the QuantiScan 1.5 software (Biosoft, UK). For determination of dissociation constants, the amount of bound myotilin constructs to F-actin in respect to free form was fitted with an equation for one site–specific binding using Prism 6 (GraphPad Software).

## Preparation of liposomes and liposome co-sedimentation assay

Liposomes were prepared from chloroform stocks of 10 mg/ml of POPC, 16:0–18:1 and 1 mg/ml of porcine brain L-α-phosphatidylinositol-4,5-bisphosphate, PI(4,5)P$_2$ (both Avanti Polar Lipids, USA). Lipid mixtures of 0% to 20% of PI(4,5)P$_2$ and 80% to 100% of POPC were first dried under a stream of nitrogen, and then vacuum dried for at least 1 h. Dried lipid mixtures were hydrated in 20 mM Tris, 200 mM NaCl, 2% glycerol, 0.5 mM DTT, pH 8.5, incubated for 10 min at room temperature, gently mixed, further incubated for 20 min, and sonicated for 2 min at room temperature in a water bath, followed by 5 freezing-thawing cycles. Using a mini-extruder (Avanti Polar Lipids), vesicles were then extruded through polycarbonate filters with 0.4 μm and 0.1 μm pore sizes, respectively. In order to avoid the presence of potential aggregates, the Ig3 domain of palladin, Ig1Ig2$^{250-444}$, and Doc2b, a positive control for PI(4,5)P$_2$ binding, were first centrifuged in a TLA-55 rotor for 30 min at 100,000 × g, 4°C, using an optima MAX-XP benchtop ultracentrifuge (Beckman Coulter Life Sciences, USA). For the assay, proteins and liposomes were mixed in polycarbonate tubes and incubated for 30 min at room temperature. Final concentrations of liposomes and proteins were 0.5 mg/ml and 10 μM, respectively. Mixtures were ultracentrifuged in a TLA-100 rotor for 40 min at 100,000 × g, 20°C. Pellets and supernatants were separated and equal volumes analyzed by SDS-PAGE. Gels were scanned and quantified by densitometry with the QuantiScan 1.5 software (Biosoft).

## NMR spectroscopy

For all NMR-based measurements uniformly $^{15}$N-labeled constructs, Ig1$^{250-344}$, Ig2$^{349-459}$, and Ig1Ig2$^{250-444}$ were expressed in *E. coli* BL21(DE3) using standard M9 minimal media. To generate protein quantities amenable to $^{1}$H-$^{15}$N HSQC based NMR studies, cell mass was generated in full media, before switching to minimal media for induction [76]. Protein purification was carried out as described above (see Protein expression and purification). Assignments for the Ig1 and Ig2 domains of myotilin were carried out previously and could be obtained from the Biological Magnetic Resonance Bank (Ig1, Entry Number: 7113; Ig2, Entry Number: 16370).

Assignments were reviewed in the original measurement conditions and were then transferred to the conditions used in this study by step gradients. For the construct Ig1Ig2$^{250-444}$, no assignments were available, and low expression yield was prohibitive to carrying out an assignment de novo. Spectra of all 3 forms were therefore simply overlaid, which enabled us to generate a partial assignment of Ig1Ig2$^{250-444}$. This was possible as the 2 domains seem to fold independently of each other and also do not show a dramatic change in shifts in the context of the full-length protein. A high affinity slowly exchanging interaction of myotilin and F-actin would lead to a combined molecular mass well above the NMR size limit. Weakly interacting species in fast exchange can, however, be measured on the free form, which retains favorable tumbling times and still yields information on the bound form, as the weighted average of the 2 species is detected. We thus used the shorter myotilin constructs displaying the weakest affinity to F-actin, as well as single domains.

The interaction of Ig1$^{250-344}$, Ig2$^{349-459}$, and Ig1Ig2$^{250-444}$ with F-actin was monitored using $^{1}$H-$^{15}$N HSQC measurements [77,78]. Various concentrations were used in these experiments and are indicated in the corresponding figure legends (S3E and S3F Fig). Both shift perturbations and changes in peak intensity due to the addition of F-actin were followed. While intensity changes were monitored as a simple ratio between the signal intensities ($I_{bound}/I_{free}$), shifts were tracked as a weighted average between proton and nitrogen shift changes in peak centres. All figures show the root of the sum of squared proton and nitrogen shifts in parts per million (ppm) values. While the proton shift is taken "as is," the nitrogen shift is divided by a factor of 5 in order to reflect the fact that dispersion in the nitrogen dimension is higher, while the achieved resolution is lower. Due to these circumstances, a diagonal shift has about 5 times higher changes in the nitrogen versus the proton dimension $shift = \sqrt{\Delta H^2 + \left(\frac{\Delta N}{5}\right)^2}$. In globular domains composed of β-sheets like Ig domains of myotilin, chemical shift changes are typically less localized and more dispersed on residues through the hydrogen-bonding network stabilizing adjacent β-strands in the β-sheet. We thus did not focus on individual residues, but rather on segments of residues, which are part of the interaction interface. Shift averages showing shift-patterns were therefore calculated by application of a rolling window average over 11 residues where a minimum of 2 values were required to obtain a value for the average. The z-scores given express the difference of the observed value (x) from the mean (μ) in terms of multiples of the standard deviation (σ), i.e., $z = \frac{x-\mu}{\sigma}$. The average shift μ was calculated as a simple average $\frac{\sum_{i=1}^{n} x}{n}$. The standard deviation σ was calculated as $\sigma = \sqrt{\frac{\sum_{i=1}^{n} (x-\mu)^2}{n-1}}$, $n$ being the number of all observed values.

NMR measurements were carried out on Bruker Avance spectrometer (USA) at 800 MHz and 298 K. Samples were prepared in 10 mM PIPES, 100 mM NaCl, 1 mM MgCl$_2$, 5% glycerol, pH 6.8, and D$_2$O was used at a final concentration of 10% as the lock solvent. The only exception is the F-actin gradient with Ig1, which was carried out in 10 mM imidazole, 50 mM KCl, 1

mM $MgCl_2$, 1 mM EGTA, pH 7.1. Measurement at a single F-actin concentration at pH 6.8 confirms that the intensity changes and shifts induced by F-actin are very similar in both conditions. Spectra were processed using NMRPipe [79]. All spectra recorded used Rance-Kay–detected sensitivity-enhanced HSQC sequences [77,78].

## Macromolecular modeling

The Ig1Ig2[220–452] models displaying hypothetical dimer architectures were prepared by manually moving 2 identical subunit copies where the Ig1 and Ig2 domains were in an extended relative orientation. The 2 subunits were oriented in a way to mimic Ig–Ig interactions, which would support dimer formation and to minimize steric clashes. These models were then used to calculate scattering profile using Crysol [80], and *P(r)* using GNOM [66], both part of the ATSAS software package [63].

For the model of myotilin:F-actin complex, the model of tandem Ig1Ig2 domains in semi-extended conformation based on SAXS data was used as a starting unbound structure [23]. Linker residues connecting myotilin Ig1 and Ig2 domains (residues 341 to 347) and N-terminal tail of actin (residues to 6) were designated as flexible. As a starting F-actin structure, an actin dimer was used (PDB 3J8I). The initial model was prepared by solvated docking implemented in Haddock 2.2 [32] using residues identified by chemical cross-linking and co-sedimentation assays as restraints. The best model with regard to Z-score was further refined using RosettaDock 2.3 [81] to yield the final myotilin:F-actin model.

## MST

MST measurements were performed on the Monolith NT.115 (NanoTemper Technologies, Germany) using fluorescently labeled DVD-actin. Purified DVD-actin was labeled using Monolith NT protein labeling kit RED-NHS (Amine Reactive) dye (NanoTemper Technologies) according to the manufacturer's instructions. The labeling reaction was performed in the supplied labeling buffer with a concentration of 20 μM DVD-actin. The labeled DVD-actin was diluted to 80 nM with the reaction buffer 20 mM HEPES, 100 mM KCl, 0.1 mM $CaCl_2$, 0.2 mM ATP, 1 mM DTT, 5% glycerol, 0.2% Tween-20, 0.5 mg/mL BSA, pH 8.0. Solutions of concentrated myotilin constructs were serially diluted in 2:1 ratio, with the same reaction buffer. Moreover, 5 μL of labeled DVD-actin was added to the 15 μL of unlabeled myotilin constructs. Reaction mixtures were incubated for 15 min at 25˚C, and approximately 5 μL of solution was loaded into Monolith NT Standard Capillaries (NanoTemper Technologies). Measurements were performed at 25˚C using 20% LED and 20% IR-laser power with 5 s/30 s/5 s laser off/on/off times. The myotilin dependent change in thermophoresis was described with a Hill function to determine the apparent dissociation constant $K_d$ of interaction from 3 independent measurements.

## F-actin depolymerization assays

Pyrene-labeled-F-actin filaments (30 μM, 25% pyrene labeled) were prepared by a 25-min incubation at room temperature in buffer A (10 mM HEPES, 50 mM KCl, 2 mM $MgCl_2$, pH 8.0) in the presence or absence of different concentrations of Ig1Ig2[250–444] (at molar ratios myotilin to actin as indicated on Fig 2E). Depolymerization of actin filaments was induced by 100× dilution with buffer A to a final concentration of 0.3 μM. Precut pipet tips were used for all manipulations of actin filaments, and care was taken to avoid filament shearing. Pyrene fluorescence was monitored at room temperature over time at an excitation of 365 nm and emission of 388 nm in a Jasco FP-6300 fluorescence spectrophotometer (Japan).

### Transfection of myoblasts and fluorescence recovery after photobleaching

C2C12 cells were grown in proliferation medium [15% FCS, 100 U/ml penicillin, 100 μg/ml streptomycin, 2 mM non-essential amino acids and 1 mM sodium pyruvate, in Dulbecco's Modified Eagle Medium (DMEM) with GlutaMAX]. Cells were trypsinized and transfected by nucleofection according to the recommendations of the manufacturer (Lonza, Cologne, Germany) with N-terminal EGFP-myotilin wild type and mutants (K354A, K359A, K354/359A, and K354/358/359A) (for details, see S3 Table). After transfection, cells were seeded on glass coverslips (WPI, Berlin, Germany) in proliferation medium. The medium was changed 24 h after transfection, and cells were allowed to differentiate at 90% confluence by changing the medium to differentiation medium (2% horse serum, 100 U/ml penicillin, 100 μg/ml streptomycin, 2 mM non-essential amino acids and 1 mM sodium pyruvate, in DMEM with Gluta-MAX). All media and supplements were from Life Technologies (USA). Seven days later, FRAP experiments were performed using a Cell Observer Spinning Disk Confocal Microscope (Carl Zeiss, Jena, Germany) equipped with an external 473-nm laser coupled via a scanner (UGA-40, Rapp OptoElectronic, Hamburg, Germany). Cells were continuously kept at 37°C and 5% $CO_2$. Zen 2012 software was used for image processing. For FRAP analysis, 8 independent experiments were performed. Regions of interest (ROIs) were limited to a single Z-disc, and for each cell, 1 to 3 ROIs were chosen and bleached, for a pulse time of 1 ms with 8 iterations, using the 473-nm laser (100 mW) with 100% intensity. A series of 3 images was taken before bleaching. Immediately after photobleaching, fluorescence recovery was monitored with an interval of 0.1 to 1 s until the signal was fully recovered (300 s). The ImageJ package Fiji was used to determine the fluorescence intensity of bleached and unbleached areas at each time point. Raw data were transformed into normalized FRAP curves as previously described [82].

### ITC

ITC experiments were performed at 25°C on a MicroCal PEAQ-ITC calorimeter (Malvern Panalytical). Protein samples were dialyzed against 20 mM Hepes, 400 mM NaCl, 250 mM arginine, 5% glycerol, pH 7.5. All solutions were degassed before the ITC experiments. Titrations consisted of sequential injections, in which 400 μM of ACTN2-EF14 was added to a sample cell containing 20 μM of either Trx-MYOT or Trx-MYOT-NEECK. The heats of dilution of ACTN2-EF14 into the buffer were determined separately and subtracted from the titration prior to analysis. All data were analyzed using MicroCal PEAQ-ITC analysis software (Malvern Panalytical).

### Pull-down assay

Before each experiment, proteins were centrifuged at 100,000 × g for 30 min, 4°C. Trx-MYOT (5 μM) was mixed with various α-actinin-2 constructs (2.5 to 7.5 μM; for details, see Fig 4F) in 0.1 ml of buffer P (20 mM Tris-HCl, 100 mM NaCl, 10 mM imidazole, 5% glycerol, pH 7.5) and incubated for 30 min at room temperature. A total of 25 μl of $Co^{2+}$-loaded Chelating Sepharose beads (Cytiva) were added to each protein mixture, followed by incubation for 10 min at room temperature. After extensive washing with buffer P, proteins bound to the beads were eluted with 50 mM Tris, 500 mM NaCl, 500 mM imidazole, 7 M urea, pH 8.0, and analyzed by SDS-PAGE.

### DSF assay

DSF was done following the protocol of [83] using a pH screen described in [84]. Briefly, a master mix of protein/dye was prepared by mixing 420 μl of 0.5 mg/ml of Trx-MYOT with

2.1 µl 5,000 × SYPRO Orange (Thermo Fisher Scientific) and filled up with 20 mM Tris, 400 mM NaCl, 250 mM arginine, 2 mM β-mercaptoethanol, pH 7.5 to 525 µl. A total of 5 µl of the protein/dye master mix was added to each well of 96-well PCR plate containing 20 µl of the individual pH screen solution (S2C Fig). The plate was sealed, tapped to mix, and centrifuged for 30 s at 1,500 × g. For the stability screen, the temperature was increased from 15˚C to 95˚C in 0.5˚C (10-s hold time) increments. A fluorescence reading was done every 0.5˚C. Data analysis was performed using the CFX Manager software (Bio-Rad, USA) included with the real-time PCR machine.

## Supporting information

**S1 Fig. Related to Fig 1. Myotilin displays conformational ensemble in solution. (A)** Determination of molecular mass across the RI/RALLS elution peak collected in parallel to SEC-SAXS mode. Molecular mass estimates are shown in red, RALLS trace in orange and RI trace in blue. Molecular mass across the main elution peak is 70 ± 7 kDa. Molecular mass estimates corresponding to elution time points prior to the main elution peak suggest a small presence of higher oligomeric species such as dimers. **(B)** Dimensionless Kratky plot indicates that while Trx-MYOT does contain ordered regions (corresponding to the Trx moiety and the Ig1 and Ig2 domains), most of the polypeptide chain is disordered and extended. In comparison, dimensionless Kratky plot of BSA, representing a typical folded protein, is shown. **(C and D)** $D_{max}$ and $R_g$ distributions from EOM analysis of Trx-MYOT. The parameters derived from the selected pool compared to the original pool suggest that Trx-MYOT is slightly restrictive in its flexibility and occupies a more compact conformation (EOM fit and models are shown in Fig 1C and 1E). **(E)** Schematic representation of $Ig1Ig2^{220-452}$ depicting potential modes of dimerization. **(F)** Theoretical $P(r)$ vs. $r$ plots calculated for potential $Ig1Ig2^{220-452}$ dimerization modes showed in (E). Of these, both parallel and antiparallel dimers using tandem Ig1Ig2 as dimerization interface (parallel$^{(via\ Ig1Ig2)}$, antiparallel$^{(via\ Ig1Ig2)}$) display marginal increase in the $D_{max}$. Staggered dimers display notable increase in the $D_{max}$ similar to experimentally observed (Fig 1F, S1 Table). In comparison to experimental data, where $Ig1Ig2^{220-452}$ adopts a conformational ensemble in solution (Fig 1E) [23], $P(r)$ vs. $r$ plots for different dimerization modes were calculated using 1 (static) conformation, consequently resulting in a longer $D_{max}$ compared to experimentally derived. In order to compare various $P(r)$ functions, $P(r)$ was normalized to the peak height. **(G)** $P(r)$ vs. $r$ plot for the concentration series of $Ig1Ig2^{220-452}$ and $Ig1Ig2^{220-452\ R405K}$ measured at the same experimental conditions (for details, see S1 Table). Inset, respective concentration series with the corresponding SAXS profiles. Note, both $Ig1Ig2^{220-452}$ and $Ig1Ig2^{220-452\ R405K}$ showed comparable concentration-dependent increase in $D_{max}$, suggesting that in vitro R405K mutation does not impair myotilin homodimerization. Data points that were used to create graphs are reported in S2 Data. EOM, ensemble optimization method; RALLS, right-angle laser light scattering; RI, refractive index; Trx, thioredoxin. (TIF)

**S2 Fig. Related to Fig 2. Myotilin binds to F-actin via Ig domains and disordered flanking regions and influences its dynamics. (A and B)** Binding of various myotilin constructs to F-actin in conditions B1 (A) or conditions B2 (B). The exponential binding curves fitted for each set of data points were used to determine apparent binding affinities shown in Fig 2A. Plotted values represent the mean ± SEM from 3 to 5 independent experiments. **(C)** Melting temperature ($T_m$) of Trx-MYOT in various buffers as assessed by DSF assay (for details, see Materials and methods). The pH of each buffer (blue rectangle) is indicated. $T_m$ max (violet) denotes the buffer conditions in which $T_m$ of Trx-MYOT was the highest (100 mM MES, pH 5.5). Brown line indicates $T_m$ in the standard buffer (100 mM Tris, pH 8.0). Note trend in increasing the

stability of Trx-MYOT by lowering the pH. **(D)** Structural analysis of the functional (F-actin binding) sites on various Ig domains. Top panel: Basic amino acid residues with surface-exposed sidechains are shown as sticks (cyan). For palladin and filamin A, this region corresponds to the F-actin binding region. Bottom panel: The functionally important residues as predicted by evolutionary coupling analysis and folding server EVfold [30] are shown as spheres and coincide with the residues shown in the top panel on the lateral sides of the Ig domains. More intense (darker) color depicts a higher probability that the residue represents a functional site. **(E)** Binding of various myotilin constructs and control (GST) to fluorescently labeled monomeric DVD-actin measured by MST. **(F)** Table of the constructs and their affinity to DVD-actin obtained from the data shown in (E). Data points that were used to create graphs are reported in S2 Data. DSF, differential scanning fluorimetry; MST, microscale thermophoresis; Trx, thioredoxin.
(TIF)

**S3 Fig. Related to Fig 3. Integrative model of myotilin:F-actin complex. (A–C)** Cross-linking of myotilin Ig1Ig2$^{250-444}$ (A), Ig1Ig2$^{185-454}$ (B), and Ig1Ig2$^{185-498}$ (C) with F-actin. Myotilin-actin complexes were cross-linked with DMTMM, either at increasing concentrations of DMTMM and fixed amount of proteins (Ig1Ig2$^{250-444}$) or at its fixed concentration and varying amounts of proteins (Ig1Ig2$^{185-454}$ and Ig1Ig2$^{250-498}$). SDS-PAGEs of cross-linked samples stained with Coomassie Brilliant Blue are shown. Specific bands, corresponding to 1:1 complex, were analyzed by MS, and data obtained from it are summarized in S2 Table. The uncropped gels can be found in S1 Data. **(D)** Model of the full-length myotilin bound to F-actin. Ig domains of myotilin were docked on F-actin and combined with flexible (not docked) models of N-terminal and carboxyl-terminal parts of the protein from EOM analysis of Trx-MYOT (Fig 1E). The 4 most populated conformations of N-terminal and carboxyl-terminal parts are shown in distinct colors. Selected residues of actin found in cross-links with myotilin are shown in orange. See also Fig 3 and S2 Table. **(E)** Relative change of $^{1}$H-$^{15}$N HSQC cross-peak intensities upon addition of F-actin to Ig1$^{250-344}$ (blue, left panel) or Ig2$^{349-459}$ (red, right panel). Values were averaged by a sliding window function over 11 amino acids. Stronger reduction in signal intensity of Ig2$^{349-459}$ as compared to Ig1$^{250-344}$ indicates that Ig2 binds more tightly to F-actin than Ig1. In all experiments, except for the one indicated by an asterisk (67 μM), a fixed concentration of myotilin was used (100 μM), while the concentration of F-actin was varied as indicated in the figure. **(F)** $^{1}$H-$^{15}$N HSQC monitored shift changes in Ig1Ig2$^{250-444}$ (50 μM) upon addition of F-actin (4 μM). Shifts were averaged by a sliding window function over 11 amino acids. In Ig1Ig2$^{250-444}$, 4 main regions with the most pronounced effect upon addition of F-actin were identified (boxed green). Inset, topology diagram of I-type Ig domain. **(G)** Shifts used in (F) expressed using z-scores (see Methods for pertinent details on calculation of z-score). For all regions indicated in (F), we find more than 1 shift with a z-score above 1 and at least 1 higher than 2. Data points that were used to create graphs are reported in S2 Data. The 3D model presented in this panel is available in the following link: https://skfb.ly/6YGRn. DMTMM, 4-(4,6-dimethoxy-1,3,5-triazin-2-yl)-4-methylmorpholiniumchloride; EOM, ensemble optimization method; HSQC, heteronuclear single quantum coherence; MS, mass spectrometry; Trx, thioredoxin.
(TIF)

**S4 Fig. Related to Fig 4. Myotilin regulates binding of tropomyosin to F-actin and does not interact with PI(4,5)P₂. (A)** Effects of myotilin on tropomyosin:F-actin interaction. Example of the SDS-PAGE used to generate Fig 4B is shown. Tpm was incubated with F-actin before addition of Trx-MYOT at molar ratios indicated on the figure. F-actin and proteins bound were sedimented by centrifugation, and equal amounts of supernatant (S) and pellet (P)

fractions were subjected to SDS-PAGE. **(B)** Effect of myotilin on α-actinin:F-actin interaction. Example of the SDS-PAGE used to generate Fig 4D is shown. ACTN2-NEECK was incubated with F-actin before addition of Tpm at molar ratios indicated on the figure. F-actin and proteins bound were sedimented by centrifugation, and equal amounts of supernatant (S) and pellet (P) fractions were subjected to SDS-PAGE. **(C)** Model of tandem Ig domains of myotilin, ACTN2 ABD, and tropomyosin bound to F-actin was generated by superposition of our myotilin:tropomyosin:F-actin model shown on Fig 4A with the structures of F-actin bound to α-actinin-2, spectrin and filamin A, and ABDs, at nominal resolutions of 16, 6.9, and 3.6 Å, respectively [38–40]. In the next step, crystal structure of α-actinin-2 ABD (salmon color, PDB: 5A36) [85] was superimposed over ABD of filamin A (homologous to ABD of cardiac and skeletal muscles expressed isoform filamin C) to obtain the final model. Dotted line (black) indicates position of tandem Ig domains of myotilin (myot Ig1, myot Ig2) on F-actin as shown on Fig 4A. N-terminal regions preceding the ABDs increase affinity to F-actin and are isoform specific [39]. The potential trace of the α-actinin-2 N-terminal extension (N-term.) on F-actin, as deduced from high-resolution structures of F-actin decorated by filamin A and spectrin ABDs is shown with a dashed red line [40]. In addition, actin-binding sites of α-actinin-2 and myotilin clearly overlap, explaining mutually exclusive binding of myotilin, α-actinin-2 and tropomyosin to F-actin. Residues of F-actin found in cross-links with N-terminal and carboxyl-terminal regions flanking myotilin Ig domains are shown in orange (see Fig 3). **(D and E)** Binding of myotilin to PI(4,5)P$_2$. Tandem of myotilin Ig domains (Ig1Ig2$^{250–444}$), Ig3 domain of palladin (Palladin Ig3), and Doc2b were incubated alone (control) or with different concentrations of PI(4,5)P$_2$ (0%–20%) in POPC vesicles and centrifuged. Supernatant (S) and pellet (P) fractions obtained were separated on SDS-PAGE. (D) Representative gels from one of the 3 independent experiments are shown. (E) Quantitative representation of (D), where the amount of protein found in pellet (protein in pellet) is represented as the percentage of the total amount of the protein found in both supernatant and pellet fractions. Mean values (± SEM) of 3 independent experiments are shown. Significance was assessed using 2-tailed Student $t$ test, $^*p < 0.05$ and $^{**}p < 0.01$. Note, myotilin does not bind to PI(4,5)P$_2$ containing vesicles. **(F)** Structural based sequence alignment of myotilin Ig1, myotilin Ig2, and palladin Ig3 domain. Residues that represent the putative PI(4,5)P$_2$ and Ins(1,4,5)P$_3$ binding sites on the Ig3 domain of palladin [48] are shown in green boxes. Conserved residues are shown in red. The numbers of the first and the last residue of the aligned sequences are given. The uncropped gels can be found in S1 Data, while data points that were used to create graphs are reported in S2 Data. ABD, actin binding domain; ACTN2 ABD, actin binding domain of α-actinin-2; ACTN2-NEECK, α-Actinin-2 NEECK; Doc2b, double C2-like domain-containing protein beta; POPC, 1-palmitoyl-2-oleoyl-sn-glycero-3-phosphocholine; Tpm, tropomyosin; Trx, thioredoxin.
(TIF)

**S1 Data. Gels presented in the study.** Uncropped gels used to generate Fig 4F and S3A–S3C, S4A, S4B, and S4D Figs.
(PDF)

**S2 Data. Excel file containing, in separate sheets, the numerical data for Figs 1C, 1F and 1G, 2C–2E, 4B, 4D, 4G, and 4H and 5B–5D and S1A–S1D and S1G, S2A–S2C and S2E, S3E–S3G and S4E Figs.**
(XLSX)

**S1 Table. SAXS data collection, analysis, and derived structural parameters.**
(PDF)

**S2 Table. List of major cross-links found between myotilin and F-actin.**
(PDF)

**S3 Table. List of constructs.**
(PDF)

## Acknowledgments

We thank the staff of the SAXS beamline at ESRF in Grenoble and EMBL-Hamburg for their
excellent support, especially Martin Schroer for his assistance with the SEC-SAXS/RALS setup.
    S. Raunser (Max Planck Institute of Molecular Physiology, Dortmund, Germany), M. Beck
(Wichita State University, USA), S. Martins (University of Vienna, Austria), and M.K. Rosen
(University of Texas Southwestern Medical Center, USA) are acknowledged for kindly provid-
ing clones and/or protein samples for human tropomyosin, Ig3 domain of human palladin,
Doc2b, and non-polymerizable *Drosophila melanogaster* DVD-actin mutant, respectively.

## Author Contributions

**Conceptualization:** Julius Kostan, Sibylle Molt, Dieter O. Fürst, Brigita Lenarčič, Kristina Dji-
    nović-Carugo.

**Data curation:** Kristina Djinović-Carugo.

**Funding acquisition:** Dmitri I. Svergun, Bettina Warscheid, Robert Konrat, Dieter O. Fürst,
    Brigita Lenarčič, Kristina Djinović-Carugo.

**Investigation:** Julius Kostan, Miha Pavšič, Vid Puž, Thomas C. Schwarz, Friedel Drepper,
    Sibylle Molt, Melissa Ann Graewert, Claudia Schreiner, Sara Sajko, Peter F. M. van der
    Ven, Adekunle Onipe.

**Methodology:** Julius Kostan, Miha Pavšič, Vid Puž, Thomas C. Schwarz, Friedel Drepper,
    Sibylle Molt, Melissa Ann Graewert, Claudia Schreiner, Sara Sajko, Peter F. M. van der
    Ven, Adekunle Onipe.

**Resources:** Dmitri I. Svergun, Bettina Warscheid, Robert Konrat, Dieter O. Fürst, Kristina Dji-
    nović-Carugo.

**Software:** Dmitri I. Svergun.

**Supervision:** Bettina Warscheid, Robert Konrat, Dieter O. Fürst, Brigita Lenarčič, Kristina
    Djinović-Carugo.

**Validation:** Julius Kostan, Dmitri I. Svergun, Bettina Warscheid, Robert Konrat, Dieter O.
    Fürst, Kristina Djinović-Carugo.

**Visualization:** Julius Kostan, Miha Pavšič, Thomas C. Schwarz, Melissa Ann Graewert, Peter
    F. M. van der Ven.

**Writing – original draft:** Julius Kostan, Miha Pavšič, Vid Puž, Thomas C. Schwarz, Friedel
    Drepper, Sibylle Molt, Melissa Ann Graewert, Peter F. M. van der Ven, Dieter O. Fürst,
    Kristina Djinović-Carugo.

**Writing – review & editing:** Julius Kostan, Miha Pavšič, Vid Puž, Thomas C. Schwarz, Friedel
    Drepper, Sibylle Molt, Melissa Ann Graewert, Claudia Schreiner, Sara Sajko, Peter F. M.
    van der Ven, Adekunle Onipe, Dmitri I. Svergun, Bettina Warscheid, Robert Konrat, Dieter
    O. Fürst, Brigita Lenarčič, Kristina Djinović-Carugo.

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
