## [Editor Report · Decision Letter 0]

22 Nov 2020

Dear Dr Djinovic-Carugo, 

Thank you for submitting the revised version of your manuscript entitled "Molecular basis of F-actin regulation and sarcomere assembly via myotilin" for consideration as a Research Article by PLOS Biology. Thank you also for your patience while we evaluated your revision.

Your manuscript has now been evaluated by the PLOS Biology editorial staff as well as by the Academic Editor and I am writing to let you know that we would like to send your submission back to the original reviewers.

However, before we can send your manuscript to reviewers, we need you to complete your submission by providing the metadata that is required for full assessment - as this is a new submission, we need you to do this again. To this end, please login to Editorial Manager where you will find the paper in the 'Submissions Needing Revisions' folder on your homepage. Please click 'Revise Submission' from the Action Links and complete all additional questions in the submission questionnaire.

Please re-submit your manuscript within two working days, i.e. by Nov 24 2020 11:59PM.

Kind regards,

Ines

--

Ines Alvarez-Garcia, PhD,

Senior Editor

PLOS Biology

---

## [Decision Letter · Decision Letter 1]

28 Jan 2021

Dear Dr Djinovic-Carugo,

Thank you for submitting your revised Research Article entitled "Molecular basis of F-actin regulation and sarcomere assembly via myotilin" for publication in PLOS Biology. Thank you also for your patience and please accept my apologies for the delay. I have now obtained advice from one of the original reviewers and have discussed these comments with the Academic Editor, who has also checked your other responses.

Based on the review (attached below), we will probably accept this manuscript for publication, assuming that you will modify the manuscript to address the remaining points raised by Reviewer 1. You will see that the reviewer still has some concerns regarding the dimerization models and the data supporting the interpretation of the structural model in general. The reviewer thinks that your claim that the model provides a platform for understanding the molecular basis of disease-causing mutations is overstated and needs to be toned down, as this is not solved in the paper. Please also make sure to address the data and other policy-related requests noted at the end of this email.

We expect to receive your revised manuscript within two weeks. Your revisions should address the specific points made by the reviewer.

-  a cover letter that should detail your responses to any editorial requests.

-  a Response to Reviewers file that provides a detailed response to the reviewer' comments

*Published Peer Review History*

*Early Version*

Sincerely,

Ines

--

Ines Alvarez-Garcia, PhD,

Senior Editor,

PLOS Biology

Fig. 1C, F, G; Fig. 2C-E; Fig. 4B, D, G, H; Fig. 5B-D; Fig. S1A-D, F, G; Fig. S2A-C, E; Fig. S3D, E, F and Fig. S4E

BLURB

Please also provide a blurb which (if accepted) will be included in our weekly and monthly Electronic Table of Contents, sent out to readers of PLOS Biology, and may be used to promote your article in social media. The blurb should be about 30-40 words long and is subject to editorial changes. It should, without exaggeration, entice people to read your manuscript. It should not be redundant with the title and should not contain acronyms or abbreviations. For examples, view our author guidelines: https://journals.plos.org/plosbiology/s/revising-your-manuscript#loc-blurb

Reviewers’ comments

Rev. 1:

As a reviewer of the previous manuscript submitted by these authors entitled "Structural insights into F-actin regulation and sarcomere assembly via myotilin," I am pleased by the significantly modified manuscript now under review. The authors have made a good effort to modify the manuscript with the suggestions from all three previous reviewers, including the removal of most results and discussion regarding evidence of dimerization. They intend to publish this part as a separate story and the greatly simplifies the complexity of their current manuscript. Yet, they still include some discussion of the dimerization data and continue to advocate for the tail-to-tail dimerization model with minimal data to support this staggered arrangement over the parallel model. While the authors have provided more rationale and potential modes of dimerization in S1E, the experimentally determined Dmax values of ~152 (Table S1) at the highest concentration don't eliminate any of the "staggered" model predicted values (159-179).

Authors do admit that the CSP values for the titration of F-actin into myotilin are "quite small," and claim to include the HSQC in Figure S3D. However, this data is an abstraction or "interpretation of patterns" of the raw data and I am not sure this is an acceptable practice in the field. Figure 3 in the text has been improved, but it should be noted that very few data restraints from NMR or "co-sedimentation" (which should be mutagenesis) inform the structural model for binding to F-actin. So few, that I question whether Figure 3B really should be included as I don't think it adds much beyond Figure 3A and is repeated to some degree again in model in Figure 6. Figure 6 legend also needs to be revised to better explain what the red and blue/purple balls are. There is also an unclosed parenthesis, too.

Most of the main claims of this paper are based on the structural model developed from several different structural studies, including SAXS, NMR, XL-MS and mutagenesis; however, I think that the authors overstate the claim that this model "provides a platform for understanding the molecular basis of disease-causing mutations…" First of all, most mutations lie in the N-terminal region which is not resolved in their model at all. The only exception is the R405K mutation which they show does not impair homodimerization or actin binding, leaving this mutation still a mystery as to effect on function.

I did approve of the changes to the summary statement and inclusion of a "statement of significance" which is accurate and highlights what I feel is the most interesting result regarding the displacement of tropomyosin by myotilin.

The data better described in the context of current literature and I feel that the manuscript will be of interest to scientists in this field like myself.

With the revisions to the text, I think this paper is much closer to being acceptable for publication. The main concerns that I still have revolve around the dimerization models and weak data to support some claims made regarding interpretation of the structural model in general.

---

## [Editor Report · Decision Letter 2]

16 Feb 2021

Dear Kristina,

On behalf of my colleagues and the Academic Editor, Laura Machesky, I am pleased to say that we can in principle offer to publish your Research Article "Molecular basis of F-actin regulation and sarcomere assembly via myotilin" in PLOS Biology, provided you address any remaining formatting and reporting issues. These will be detailed in an email that will follow this letter and that you will usually receive within 2-3 business days, during which time no action is required from you. Please note that we will not be able to formally accept your manuscript and schedule it for publication until you have made the required changes.

PRESS

Thank you again for supporting Open Access publishing. We look forward to publishing your paper in PLOS Biology. 

Sincerely, 

Ines

--

Ines Alvarez-Garcia, PhD 

Senior Editor 

PLOS Biology
